# RULE-BASED POLICY REGULARIZATION FOR REINFORCEMENT LEARNING-BASED BUILDING CONTROL

## ABSTRACT

Rule-based control (RBC) is widely adopted in buildings due to its stability and robustness. It resembles a behavior cloning methodology refined by human expertise. However, it is unlikely for RBC to exceed a reinforcement learning (RL) agent's performance as deep RL model constantly evolves and is scalable. In this paper, we explore how to incorporate rule-based control into reinforcement learning to learn a more robust policy in both online and offline settings with a unified approach. We start with state-of-the-art online and offline RL methods, TD3 and TD3+BC, then improve on them using a dynamically weighted actor loss function to selectively choose which policy RL models to learn from at each time step of training. With experiments across various weather conditions in both deterministic and stochastic scenarios, we empirically demonstrate that our rule-based incorporated control regularization (RUBICON) method outperforms representative baseline methods in offline settings by $40.7\%$ and by $49.7\%$ in online settings for building-RL environments. We open-source our codes, baselines, and data for both RL and building researchers to explore the opportunity to apply offline RL in building domain.

## 1 INTRODUCTION

Most buildings implement rule-based control via building management systems, adjusting the setpoints of actuators to co-optimize occupants' thermal comfort and energy efficiency. These rule-based control systems codify the problem-solving know-how of human experts, akin to behavioral cloning policy learnt from expert demonstration without randomness and uncertainty (Hayes-Roth, 1985). While stable, such control lacks the flexibility to evolve over time.

Much research has demonstrated that RL can outperform RBC in both online and offline settings. Zhang et al. (2019) developed a framework for whole building HVAC (heating, ventilation, air-conditioning) control in online settings. Liu et al. (2022) incorporated Kullback-Leibler (KL) divergence constraint during training of an offline RL agent for stability, and deployed the policy in a real building. We focus on improving upon RL algorithms for HVAC control in buildings where a rule-based policy already exists. We use established building-RL simulation environments for our experiments, in both online and offline settings (Jiménez-Raboso et al., 2021).

Reinforcement learning is traditionally studied as an online paradigm. However, in real-world problems, configuring an accurate simulator or training environment dynamic models might be infeasible or time-consuming. Batch reinforcement learning (BRL, also known as offline reinforcement learning) learns policies using only historical data without simulators to interact with during training. RL regularization methods are typically tailored specifically to online or offline setting. For example, online methods encourage exploration to either improve the estimate of non-greedy actions' values or explore a better policy (Haarnoja et al., 2018; Ziebart et al., 2008; Haarnoja et al., 2017). On the other hand, offline methods favor exploitation since it is unlikely for BRL models to accurately estimate uncharted state-action values with a static dataset (Fujimoto et al., 2019; Wu et al., 2019).

In our work, we explore the questions: *Can we incorporate an existing rule-based control policy into training of a reinforcement learning policy to improve performance? Can this method be implemented in both online and offline settings?* TD3+BC (Fujimoto & Gu, 2021) makes minimal changes to convert an online method TD3 (Fujimoto et al., 2018) to offline mode with comparable performance as state-of-the-art BRL methods only by adding a behavior cloning term to regularize

the policy. In TD3+BC, learning behavioral policy relies on historical data. We build on this idea to regularize RL policy using an existing RBC policy combined with the behavioral policy. Our method can be incorporated into existing actor-critic RL algorithms with minimal changes.

RUBICON considers RBC as a safe policy based on which RL training can be improved. The actor selectively trains on either RBC or behavioral policy, depending on which policy yields a higher averaged Q-value in a mini-batch estimated by the critic network. Our proposed approach is distinct from prior works in three aspects: (1) We develop a unified regularization approach for both online and offline RL methods with minimal algorithmic modification. (2) Rule-based control policy is directly incorporated into the policy update step to provide stability and robustness. (3) We introduce a dynamic weighting method in actor-critic settings. The actor loss is varied from time step to time step depending on the Q-value estimate of behavioral policy and RBC policy predicted from the value networks. We empirically demonstrate that our method outperforms state-of-the-art methods in offline settings and improves on TD3 in online training in HVAC control environments. To our knowledge, previously RBC is only used as hard constraints or heuristics in RL settings, and we are the first to incorporate an existing reference policy directly into actor-critic algorithms.

## 2 RELATED WORK

**Rule-based systems** Rule-based system is one of the first artificial intelligence (AI) methods that solves many real-world problems. *Planner* (Hewitt, 1971) is a problem-solving language that embeds real-world knowledge in procedures. It was used in robotic control and as a semantic base for English. MYCIN (Shortliffe, 1974) is a rule-based problem-solving system to assist physicians with an appropriate therapy for bacterial infections. XCON, a production-rule-based system, is used to validate the technical correctness (configurability) of customer orders and to guide the actual assembly of these orders. It has about $80K$ rules and achieved $95\sim98\%$ accuracy. It saved \$25M/year by reducing errors in orders, and gained increasing customer satisfaction (Kraft, 1984).

**RL + RBC** The combination of RL and RBC has been explored in many studies, where RBCs are primarily used as auxiliary constraints or guiding mechanism. Lee et al. (2020) propose to use two modules in their control flow, one for continuous control with RL agent and a discrete one controlled by RBC. Wang et al. (2019) improve RL with low level rule-based trajectory modification to achieve a safe and efficient lane-change behavior. Zhu et al. (2021) incorporate RBC for generating the closed-loop trajectory and reducing the exploration space for RL pre-processing. Berenji (1992) use a learning process to fine-tune the performance of a rule-based controller. Radaideh & Shirvan (2021) first train RL proximal policy optimization (PPO) (Schulman et al., 2017) agents to master matching some of the problem rules and constraints, then RL is used to inject experiences to guide various evolutionary/stochastic algorithms. Likmeta et al. (2020) learn RBC parameters via RL methods. These previous methods incorporate RBC in the flow as heuristic or as hard constraints. Instead, we directly incorporate RBC policy in RL training in an algorithmic way.

**Online RL regularization** The online baseline we will compare to in evaluation is a state-of-the-art algorithm: TD3. It applies target policy smoothing regularization to avoid overfitting in the value estimate with deterministic policies. TRPO (Schulman et al., 2015) uses a trust region constraint based on KL-divergence between old and new policy distributions for robust policy updates. SAC (Haarnoja et al., 2018) uses soft policy iteration for learning optimal maximum entropy policies. Munchausen-RL (Vieillard et al., 2020) regularizes policy updates with a KL-divergence penalty similar to TRPO, and adds a scaled entropy term to penalize policy that is far from uniform policy. Our method differs from these methods in that we incorporate an existing real-world policy for online RL training.

**Offline RL regularization** Offline RL is more conservative compared with online methods as it does not require interaction with the environment. It suffers from extrapolation errors induced by selecting out-of-distribution actions. Since offline RL policies are learnt entirely from a static dataset, it is unlikely for value networks to accurately estimate the values in regime where there is no sufficient state-action visitation. Thus, regularization methods become more prominent in offline settings. Batch-constrained deep Q-learning (BCQ) (Fujimoto et al., 2019), one of the pioneers of offline RL, ascribes extrapolation errors to three main factors: absent data, model bias, and training mismatch. It mitigates the errors by deploying a variational autoencoder (VAE) to reconstruct the action given a state using the data collected by the behavioral policy. The offline baseline method we will

compare to in our study is TD3+BC. It starts from online method TD3, adds a behavior cloning term in the policy update to regularize the actor to imitate the behavioral policy and avoid selecting out-of-distribution actions. BRAC (Wu et al., 2019) studies both value penalty and policy regularization with multiple divergence metrics (KL, maximum mean discrepancy (MMD), and Wasserstein) to regularize the actor's policy based on the behavioral policy. FisherBRC (Kostrikov et al., 2021) incorporates a gradient penalty regularizer for the state-action value offset term and demonstrates the equivalence to Fisher divergence regularization. CQL (Kumar et al., 2020) learns a conservative, lower-bound estimate in value network via regularizing Q-values. Model-based method, e.g. COMBO (Yu et al., 2021) regularizes the value function on out-of-support transitions generated via environment dynamic models' rollouts.

All of these prior works use data collected by a behavioral policy, and do not assume access to any existing policy. The behavioral policy used in experiments is typically a random agent or an RL agent trained partially (medium agent) or to convergence (expert agent). In contrast, we assume direct access to a robust behavioral policy in the form of rule-based control. While this assumption may not hold for robotic applications typically used to evaluate offline RL algorithms, rule-based control policies are routinely deployed in industrial control settings, such as building HVAC control.

The regularization methods in both online and offline settings regularize with ensemble models, divergence between partially trained policies, or with logged data. These algorithms still fall into the deadly triad of function approximation, bootstrapping, and off-policy training (Sutton & Barto, 2018). In our case, we incorporate a robust reference policy to improve RL policy performance. The behavioral policy is selectively trained with actor only when the averaged Q-value estimate is higher than the rule-based control policy actions in every mini-batch. The rule-based control policy reduces uncertainty due to its deterministic behavior. On the opposite, deep learning model is affected by random initialization conditions, even if trained on the same dataset, as varied initialization conditions might lead to different policies.

## 3 BACKGROUND

In the reinforcement learning paradigm, an agent acts in an environment, sequentially selects actions based on its policy at every time step. The problem can be formulated as a Markov Decision Process (MDP) defined by a tuple $(\mathcal{S}, \mathcal{A}, \mathcal{R}, p, \gamma)$, with state space $\mathcal{S}$, action space $\mathcal{A}$, reward function $\mathcal{R}$, transition dynamics $p$, and discount factor $\gamma \in [0, 1)$. The goal is to maximize the expectation of the cumulative discounted rewards, denoted by $R_t = \sum_{i=t+1}^{\infty} \gamma^i r(s_i, a_i, s_{i+1})$( Sutton & Barto (2018)). The agent's behavior is determined by a policy $\pi : \mathcal{S} \to \mathcal{A}$, which maps states to actions either in deterministic way or with a probability distribution. The expected return following the policy from a given state $s$ is the action-value function $Q^\pi(s, a) = \mathbb{E}_\pi[\sum_{t=0}^{\infty} \gamma^t R_{t+1} | s_0 = s, a_0 = a]$ by taking action $a$.

We use the building RL environment from Jiménez-Raboso et al. (2021). The objective of the agent is to maintain a comfortable thermal environment with minimum energy use. The state consists of indoor/outdoor temperatures, time/day, occupant count, thermal comfort, and related sensor data. The action adjusts the temperature setting of the thermostat. The reward is a linear combination of occupants' thermal comfort and energy consumption. The environment is a single floor building divided in 5 zones, with 1 interior and 4 exterior rooms. See Appendix B for details about our building RL settings.

## 4 RULE-BASED INCORPORATED CONTROL REGULARIZATION

Our goal is to improve an agent's ability to learn with the assistance of human expert's domain knowledge in both online and offline settings. In real-world problems, sometimes we have existing simulators as oracles so we can safely learn with online RL methods before deploying in the real environment, for example, in robot control (Todorov et al., 2012), Go (Silver et al., 2017), and video games (Mnih et al., 2013). However, in some real-world problems, it is time-consuming and requires domain expertise to build a functional simulation environment (e.g. building thermal simulations), or it can be dangerous or risky to evaluate partially trained policy (e.g. healthcare and financial trading). Offline RL algorithms, on the other hand, rely on historical data collected by an existing but unknown behavioral policy. The objective is to learn a policy that improves on the behavioral

policy measured through episodic rewards. In Fig. 1, we illustrate the process of RL training that accommodates both the online and offline paradigms.

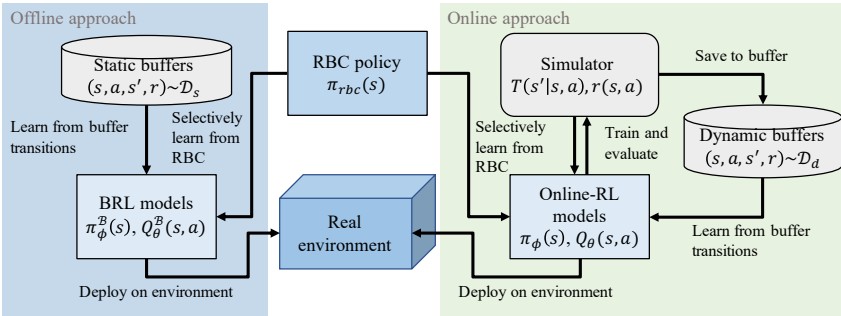

Figure 1: Our proposed method, RUBICON, it incorporates RBC into RL to improve stability in building HVAC control. It could be applied on both online and offline approaches.

Our algorithm builds on existing actor-critic algorithms TD3 and TD3+BC. We only modify the policy update strategy with the rule-based control policy, and use the critic as-is. Therefore, we focus our discussion on the policy update of the algorithm. TD3 starts from DDPG (Silver et al., 2014) and mitigates the function approximation error with double Q-learning and delayed policy updates. TD3+BC is an offline RL algorithm adapted from TD3, and is one of the state-of-the-art offline RL methods evaluated with D4RL datasets (Fu et al., 2020). TD3+BC adds a behavior cloning term to the policy update step to penalize the policy that is far away from behavioral policy (Eq. 1). The blue-colored terms indicate the changes from TD3 to TD3+BC.

$$\pi = \arg\max_{\pi} \mathbb{E}_{(s,a)\sim\mathcal{D}} \left[ \lambda Q(s, \pi(s)) - (\pi(s) - a)^2 \right] \tag{1}$$

$$\lambda = \frac{\alpha}{\frac{1}{N}\sum_{(s_i,a_i)}|Q(s_i,a_i)|} \tag{2}$$

In Eq. 1, $\lambda$ is decided by the averaged mini-batch Q-estimate and a hyperaparameter $\alpha$ to adjust between RL and imitation learning (Eq. 2).

Our method, RUBICON, dynamically weighs both TD3 and TD3+BC's policy update steps with either RBC policy or behavioral policy in each training iteration. In Eq. 3, we replace the actions $a$ sampled from the buffers in Eq. 1 with $\pi_{Q_{max}}(s)$ and add a hyperparameter $\xi$ to integrate TD3 and TD3+BC methods as one. Red-colored terms indicate the changes from TD3 and TD3+BC to our method. We replace the notation of sampled actions $a$ in TD3+BC with behavioral policy $\pi_b(s)$ to avoid confusion. Details of the hyperparameter settings in our work are in Appendix D.

$$\pi = \arg\max_{\pi} \mathbb{E}_{s\sim\mathcal{D}} \left[ \lambda Q(s, \pi(s)) - \xi(\pi(s) - \pi_{Q_{max}}(s))^2 \right] \tag{3}$$

$$\pi_{Q_{max}}(s) = \arg\max_{\pi}\{\bar{Q}(s, \pi_b(s)), \bar{Q}(s, \pi_{rbc}(s)\} \tag{4}$$

Every time when the policy is being updated, given the states $s$ of the sampled mini-batch, the behavioral policy $\pi_b(s)$ and the RBC policy $\pi_{rbc}(s)$ select actions in a deterministic fashion. The state-action pairs' Q-values are estimated by the critic, the average of the Q-value estimations in the mini-batches are $\bar{Q}(s, \pi_b(s))$ and $\bar{Q}(s, \pi_{rbc}(s))$. We dynamically select the transitions with a higher average Q-value to be regularized from in each policy update step, i.e. the actor loss function is dynamically weighted. In online settings, the behavioral policies are the older versions of the policy we used to generate the buffer.

The reason we choose the average as the metric to decide which set of transitions to learn from instead of selecting each transition with higher estimated value (each batch is a combination of $\pi_b(s)$ and $\pi_{rbc}(s)$ ) is that if we choose by each transition we will lose the information on which state-action visitations lead to worse values, the model will then suffer from the imbalanced data problem.

The credit/blame assignment is essential in RL learning convergence and the experience replay can help speed up the propagation process (Lin, 1992). The algorithm of our method is given in Alg. 1. Changes from baselines to our method are highlighted in blue. $d$ is the policy update frequency, the noise $\epsilon$ added to the policy is sampled with Gaussian $\mathcal{N}(0, \sigma)$ and clipped by $c$. In both online and offline approaches, the policy update follows Eq. 3 and 4 with different hyperparameter settings.

Our rule-based control algorithm is described in Alg. 2. It is derived from the rule-based controller in Sinergym's (Jiménez-Raboso et al., 2021) example. For the purpose of computation efficiency and to fit the batch settings in our algorithm, we vectorize the original RBC policy. The rules are simple and intuitive, and can generalize well: First, we get the datetime information we need from the states. Then, we can get the seasonal comfort temperature zone for every transition. If the indoor air temperature (IAT) is below the lower bound of the comfort zone, then we set both cooling and heating setpoints a degree higher (measured in Celcius degree). On the opposite, if the IAT is above the upper bound of the comfort zone, then we set both the heating and cooling setpoints a degree lower than the current setpoints. Finally, we examine if the current datetime is in the office hours. If not, then the setpoints are set to be $(18.33, 23.33)$ (°C) for the purpose of energy reduction since occupants' thermal comfort is not important in these time periods assuming zero occupancy.

---

**Algorithm 1:** RUBICON

Initialize critic networks $Q_{\theta_1}, Q_{\theta_2}$, actor network $\pi_\phi$, and RBC policy $\pi_{rbc}$ with random parameters $\theta_1, \theta_2, \phi$, target networks $\theta'_1 \leftarrow \theta_1, \theta'_2 \leftarrow \theta_2$, $\phi' \leftarrow \phi$ and replay buffer or load replay buffer $\mathcal{B}$

**for** $t = 1$ *to T* **do**
  **if** *online* **then**
    Select action with exploration noise
      $a \sim \pi_\phi(s) + \epsilon, \epsilon \sim \mathcal{N}(0, \sigma)$
    Observe reward $r$ and next state $s'$
    Store transition $(s, a, r, s')$ in $\mathcal{B}$
  Sample mini-batch of $N$ transitions $(s, a, r, s')$
    from $\mathcal{B}$
  $\tilde{a} \leftarrow \pi_{\phi'}(s') + \epsilon, \epsilon \sim \text{clip}(\mathcal{N}(0, \tilde{\sigma}), \text{-c,c})$
  $y \leftarrow r + \gamma \min_{j=1,2} Q_{\theta'_j}(s', \tilde{a})$
  Update critics
    $\theta_j \leftarrow \arg\min_{\theta_j} N^{-1}(y - Q_{\theta_j}(s, a))^2$
  **if** $t \bmod d$ **then**
    Update $\phi$ by policy gradient:
    Policy update follows Eq. 3 and 4
    Calculate $\nabla_\phi J(\phi)$
    Update target networks:
    $\theta'_j \leftarrow \tau\theta_j + (1 - \tau)\theta'_j$
    $\phi' \leftarrow \tau\phi + (1 - \tau)\phi'$

**Algorithm 2:** Rule-based control policy

**Input** : Current daytime $current\_dt$, indoor air temperature $IAT$, zone thermostat heating setpoint temperature $a_h$, and zone thermostat cooling setpoint temperature $a_c$, all obtained from the states in the sampled mini-batch with size $N$

**Output:** Actions selected by RBC

**for** $i$ *in N* **do**
  $season\_comfort\_zone_i =$
  $get\_season\_comfort(current\_dt_i)$
  **if** $IAT_i >= min(season\_comfort\_zone_i)$
  **then**
    $a_{h_i} = a_{h_i} - 1$
    $a_{c_i} = a_{c_i} - 1$
  **if** $IAT_i < max(season\_comfort\_zone_i)$
  **then**
    $a_{h_i} = a_{h_i} + 1$
    $a_{c_i} = a_{c_i} + 1$
  $a_i = (a_{hi}, a_{ci})$
  **if** $current\_dt_i.weekday \geq 5$ *or*
  $current\_dt_i.hour\ in\ range(22,6)$ **then**
    $a_i = (18.33, 23.33)(°C)$

---

## 5 EXPERIMENTS

In our experiments, there are two environment types: deterministic and stochastic. A Gaussian noise with $\mu=0$ and $\sigma=2.5$ is added to the outside temperature from episode to episode in the stochastic environments. And three weather types: hot, cool, and mixed. In the results, "*hot-deterministic*" indicates that the task is learnt and evaluated with hot weather condition and deterministic environment. Similarly, we have all six combinations such as "*cool-stochastic*" and so on. More details about the RL setup is given in Appendix B. All scores in tables and figures in this paper are normalized with expert policy as 100 and random policy as 0.

### 5.1 OFFLINE APPROACH

First, we consider the offline approach, where no simulator exists but historical data is available. We follow the standard procedure for BRL evaluation (Fu et al., 2020): (1.) Train behavioral agents for 500K time steps, then compare the most representative algorithms DDPG, TD3, and SAC (learning curves are shown in Appendix C). The online methods we compare are described below:

- **DDPG**: Deep deterministic policy gradient is a method that combines the actor-critic approach and deep Q-network (DQN) (Mnih et al., 2013). It is capable of dealing with continuous action spaces problems via policy gradient in a deterministic approach which outperforms stochastic policy method in high dimensional tasks.

- **SAC**: Soft actor-critic, an off-policy maximum entropy RL algorithm to encourage exploration. They empirically show that SAC yields a better sample efficiency than DDPG.

- **TD3**: Twin delayed deep deterministic policy gradient algorithm, it reduces overestimation with double Q-learning, combines with target networks to limit errors from imprecise function approximation.

(2.) Select the best agent as our expert agent and generate buffers with it for 500K time steps. A medium agent is trained "halfway", it means that an agent is trained most closed to an agent with the evaluation performance half the performance as the expert agent. And a random agent which samples actions randomly and generate buffers.

(3.) Train BRL methods for 500K time steps and evaluate the policy every 25K time steps in all buffers mentioned above in step (2.). We show the detailed learning curves in Appendix C. Normalized and averaged scores across runs are shown in Table 1. The offline methods we compare with are listed below:

- **TD3+BC**: An offline version of TD3, it adds a behavior cloning term to regularize policy towards behavioral policy combined with mini-batch Q-values and buffer states normalization for stability improvement.

- **CQL**: Conservative Q-learning, derived from SAC, it learns a lower-bound estimate of the value function by regularizing the Q-values during training.

- **BCQ**: Batch-constrained deep Q-learning, it implements a variational autoencoder (VAE) (Kingma & Welling, 2013) to reconstruct the action given the state. And adds perturbation in actor on the policy, the degree of perturbation and size of mini-batch can be adjusted in order to behave more like a traditional RL method or imitation learning.

- **BC**: Behavior cloning, we train a VAE to reconstruct action given state. It simply imitates the behavioral agent without reward signals.

In Table 1, we observe that RUBICON outperforms all other benchmarks in overall score across weather types, random seeds, and environment types. Other BRL methods show good performance either in specific tasks or with a specific randomly initialized configuration; however, overall they are more unstable cf. RUBICON. Our method provides more robust and consistent performance across all variants and demonstrates the ability to generalize across various weather types and response modes of tasks. Also, as we can see in Fig. 2, learning from both medium buffer and RBC policy, RUBICON improves on their best performance. Our method is stable since the standard deviation is the least among the policies trained. We include the BRL learning curves with expert and random buffers in Appendix C

### 5.1.1 DATA EFFICIENCY EXPERIMENT

We conduct the experiments with buffers of only one year of data ($35,040$ transitions). Data efficiency is a challenge for RL to yield accurate value estimation. In Table 2, we observe that our method still outperforms its baseline overall. Although it dominates with random buffers and has comparable performance with expert buffers, it does not learn well with medium buffers. The root cause is the similarity of the quality of actions between medium buffers and RBC policy, which causes the critic to misjudge which action to pick between them. However, RUBICON still outperforms the baseline in other two types of buffers since the value estimation differences between $(\pi_b(s), s)$ and $(\pi_{rbc}(s), s)$ are more prominent in these scenarios.

---

[1]Some scores with a standard deviation of 0 is caused by the round down of normalized scores, they are negligible numbers.

| Environment | Buffer | RUBICON | TD3+BC | CQL | BCQ | BC |
|---|---|---|---|---|---|---|
| hot-deterministic | Expert | 86.13±17.83 | 99.72±0.42 | **100±0** | -32.01±95.46 | -89.2±14.84 |
| hot-deterministic | Medium | 64.91±18.02 | -49.58±13.52 | **67.64±32.83** | 13.4±51.24 | -26.74±26.47 |
| hot-deterministic | Random | 62.7±14.36 | -45.73±44.8 | -23.19±76.76 | **69.2±33.61** | -12.55±74.63 |
| mixed-deterministic | Expert | 81±25.94 | 94.66±7.36 | **100±0** | -6.22±84.27 | -95.46±14.78 |
| mixed-deterministic | Medium | **86.84±12.39** | 36.23±56.33 | 37.36±86.8 | 64.45±37.4 | -27.82±63.16 |
| mixed-deterministic | Random | **68.83±4.93** | -13.71±57.06 | -23.46±83.61 | -65.29±48.84 | -103.4±7.45 |
| cool-deterministic | Expert | 98±2.78 | 81.11±16.88 | **100±0** | -29.74±95.89 | 27.76±102.15 |
| cool-deterministic | Medium | **72.2±8.07** | -49.97±36.4 | 55.44±49 | 70.18±14.42 | 8.62±45.32 |
| cool-deterministic | Random | **66.5±0** | -58.4±19.25 | 12.98±73.04 | 27.77±63.67 | 10.48±70.79 |
| hot-stochastic | Expert | 99.01±0.56 | 77.69±30.48 | **99.49±0.24** | -15.34±84.51 | -72.86±38.25 |
| hot-stochastic | Medium | **59.72±5.29** | -14.84±66.04 | 39.92±56.67 | -62.2±5.25 | 31.22±66.26 |
| hot-stochastic | Random | **68.83±21.26** | -1.82±73.31 | 36.64±67.61 | -1.23±68.86 | -10.45±61.91 |
| mixed-stochastic | Expert | 94.16±8.12 | 96.6±2.14 | **99.77±0.21** | -108.38±2.83 | -102.02±9.26 |
| mixed-stochastic | Medium | **87.23±12.34** | 9.48±81.06 | 80.13±20.78 | 70.75±9.9 | 38.66±48.02 |
| mixed-stochastic | Random | 67.03±6.26 | 28.01±72.79 | **94.04±5.87** | -109.46±0.77 | -107.41±4.36 |
| cool-stochastic | Expert | 53.58±65.53 | 78.27±31.08 | **99.97±0.32** | -115.85±0.98 | 28.15±101.52 |
| cool-stochastic | Medium | 68.07±0.46 | 16.09±69.41 | **81.56±18.01** | -11.55±56.13 | 25.44±35.57 |
| cool-stochastic | Random | **67.55±1.14** | -44.33±36.36 | -97.35±11.07 | -53.92±78.06 | -50.37±83.99 |
| Sum | | **1352.37±225.38** | 339.49±714.77 | 960.98±582.88 | -295.48±832.17 | -527.93±868.81 |

Table 1: BRL methods benchmark: Average normalized score over the final 5 evaluations and 3 random seeds. ± corresponds to standard deviation over the last 5 evaluations across runs. [1]

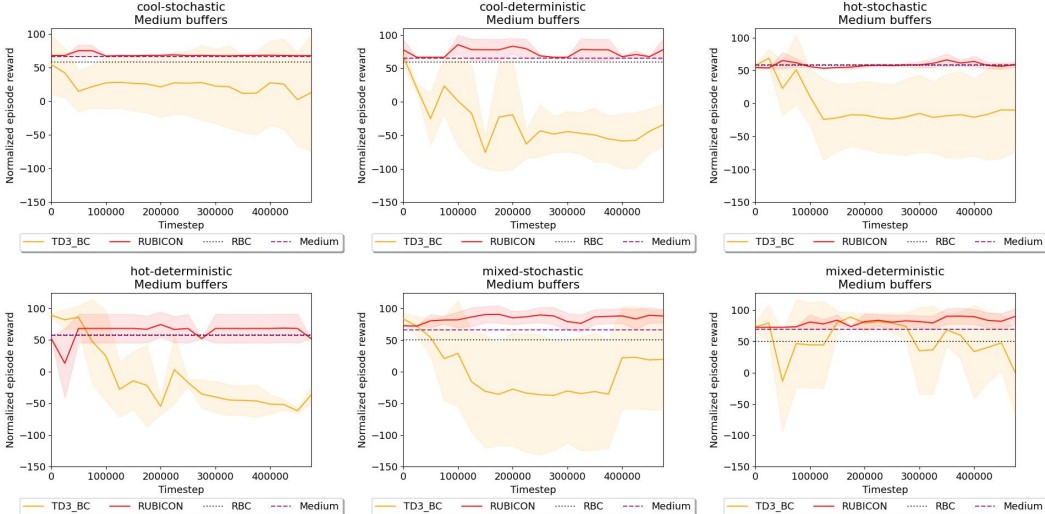

Figure 2: Learning curves of RUBICON and the baseline method TD3+BC with medium buffers

### 5.1.2 ABLATION EXPERIMENT

In this experiment, we remove the dynamically weighted regularization. Instead we regularize the behavioral policy and RBC policy simultaneously in every iteration of training (see Eq. 5). The experimental results are shown in Table 3. We observe regularizing both at the same time deteriorates policy training. Since in each iteration, one of RBC policy $\pi_{rbc}(s)$ and behavioral policy $\pi_b(s)$ yields a better action selection compared to the other. It emphasizes the necessity of dynamic weighting in the policy update steps.

$$\pi = \arg\max_{\pi} \mathbb{E}_{(s,a)\sim\mathcal{D}} \left[ \lambda Q(s, \pi(s)) - (\pi(s) - a)^2 - (\pi(s) - \pi_{rbc}(s))^2 \right] \tag{5}$$

### 5.1.3 TRANSFER EXPERIMENT

We consider a realistic scenario where in one weather condition we already have an existing buffer then we want to use it as prior knowledge to combine with RBC and transfer our model to another weather type where we have no data. We experiment with the medium buffers in stochastic environments. The results shown in Table 4 indicate that our method is capable of transferring from one weather condition to another with comparable performance without any hyperparameter changes. As the results indicate, due to the diversity of the mixed weather, it improves the learning in cool

| Environment | Buffer | RUBICON | TD3+BC |
|---|---|---|---|
| hot-deterministic | Expert | **89.65±9.12** | 74.96±12.38 |
| hot-deterministic | Medium | -6.92±78.43 | **79.37±25.5** |
| hot-deterministic | Random | **90.41±5.47** | -24.18±59.62 |
| mixed-deterministic | Expert | 43.09±60.36 | **93.77±8.71** |
| mixed-deterministic | Medium | 48.24±54.32 | **98.67±2.28** |
| mixed-deterministic | Random | **82.89±10.19** | 72.58±6.91 |
| cool-deterministic | Expert | **89.52±11.28** | 86.45±8.62 |
| cool-deterministic | Medium | -11.56±62.82 | **47.61±17.66** |
| cool-deterministic | Random | 35.92±43.24 | **42.59±72.43** |
| hot-stochastic | Expert | 80.35±25.8 | **89.02±5.51** |
| hot-stochastic | Medium | 5.3±43.33 | **10.79±56.47** |
| hot-stochastic | Random | **62.26±18.59** | 26.43±52.33 |
| mixed-stochastic | Expert | 78.08±25.78 | **85.25±12.96** |
| mixed-stochastic | Medium | 66.02±17.68 | **72.17±16.14** |
| mixed-stochastic | Random | **55.26±29.19** | -86.21±25.92 |
| cool-stochastic | Expert | **98.18±2.56** | 39.04±86.35 |
| cool-stochastic | Medium | **73.16±19.56** | 38.68±39.93 |
| cool-stochastic | Random | **69.76±5.46** | -49.5±9.75 |
| Sum | . | **1049.61±523.18** | 797.49±519.47 |

Table 2: Data reduction experiment

| Environment | Buffer | RUBICON | RUBICON w/o DW |
|---|---|---|---|
| hot-deterministic | Expert | **86.13±17.83** | -19.8±63.89 |
| hot-deterministic | Medium | **64.91±18.02** | 47.26±12.89 |
| hot-deterministic | Random | **62.7±14.36** | -19.8±63.89 |
| mixed-deterministic | Expert | **81±25.94** | -75.6±29.46 |
| mixed-deterministic | Medium | **86.84±12.39** | 42.99±48.04 |
| mixed-deterministic | Random | **68.83±4.93** | -75.6±29.46 |
| cool-deterministic | Expert | **98±2.78** | 41.3±20.53 |
| cool-deterministic | Medium | **72.2±8.07** | 36.54±67.84 |
| cool-deterministic | Random | **66.5±0** | 41.3±20.53 |
| hot-stochastic | Expert | **99.01±0.56** | 57.68±22.3 |
| hot-stochastic | Medium | **59.72±5.29** | 29.26±45.5 |
| hot-stochastic | Random | **68.83±21.26** | 57.68±22.3 |
| mixed-stochastic | Expert | **94.16±8.12** | 40.57±44.91 |
| mixed-stochastic | Medium | **87.23±12.34** | 55.6±33.53 |
| mixed-stochastic | Random | **67.03±6.26** | 40.57±44.91 |
| cool-stochastic | Expert | **53.58±65.53** | -68.84±27.46 |
| cool-stochastic | Medium | **68.07±0.46** | 8.01±61.1 |
| cool-stochastic | Random | **67.55±1.14** | -68.84±27.46 |
| Sum | | **1352.37±225.38** | 170.3±686.1 |

Table 3: Ablation experiment 1

| Environment | Trans. from | RUBICON_Trans. | RUBICON |
|---|---|---|---|
| hot-stochastic | cool-stochastic | **71.42±3.1** | 59.72±5.29 |
| mixed-stochastic | cool-stochastic | 84.93±16.92 | **87.23±12.34** |
| cool-stochastic | hot-stochastic | 54.07±0.85 | **68.07±0.46** |
| mixed-stochastic | hot-stochastic | 81.02±20.09 | **87.23±12.34** |
| cool-stochastic | mixed-stochastic | **72.39±0.68** | 68.07±0.46 |
| hot-stochastic | mixed-stochastic | **75.26±3.3** | 59.72±5.29 |
| Sum | | **439.09±44.94** | 430.04±36.18 |

Table 4: Transfer experiment

and hot weathers. On the other hand, transfer from monotonic weather condition leads to worse returns.

### 5.1.4 REWARD ANALYSIS EXPERIMENT

Since Q-value estimation is usually overestimated, we want to use immediate reward as a reference to examine the quality of the predictions of Q-functions. We pre-train a reward model $R_\psi(s, a)$ to predict reward $\hat{r}$ given state $s$ and selected action $a$ with 200K iterations with the buffer as our training data. At each iteration of policy update, we record the policy $\pi(s)$ and the predict rewards in each batch $\hat{r} = R_\psi(s, \pi(s))$. We plot the distributions of reward in action spaces in Fig. 3. It demonstrates that RUBICON selects the actions in a wider range cf. TD3+BC, however, with a reward distribution of higher values. The distribution shown is with 10% of data randomly selected from the entire training for a better visualization.

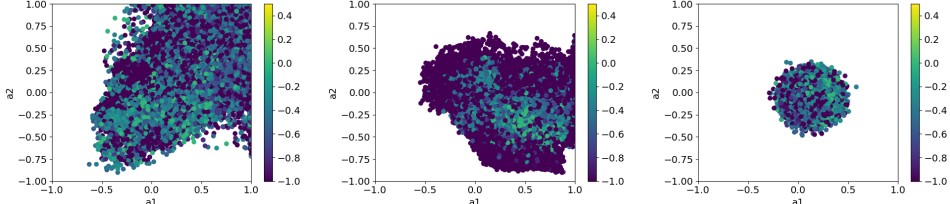

Figure 3: Reward distribution in action spaces of *hot-continuous* environment learns from medium buffer, from left to right: RUBICON (1.842/1.978/-0.577), TD3+BC (1.534/1.332/-0.668), and buffer (0.908/0.915/-0.799), tuples indicate the (a1 range/a2 range/reward mean).

## 5.2 ONLINE APPROACH

In online approach, we assume an oracle exists for accurate simulation. In real-world applications, researchers train online model in simulation before deployment in real building environments. We compare with TD3, the baseline we develop our method on. Experimental results comparing TD3 and our method can be found in Table 5. In five out of six tasks, our method outperforms TD3

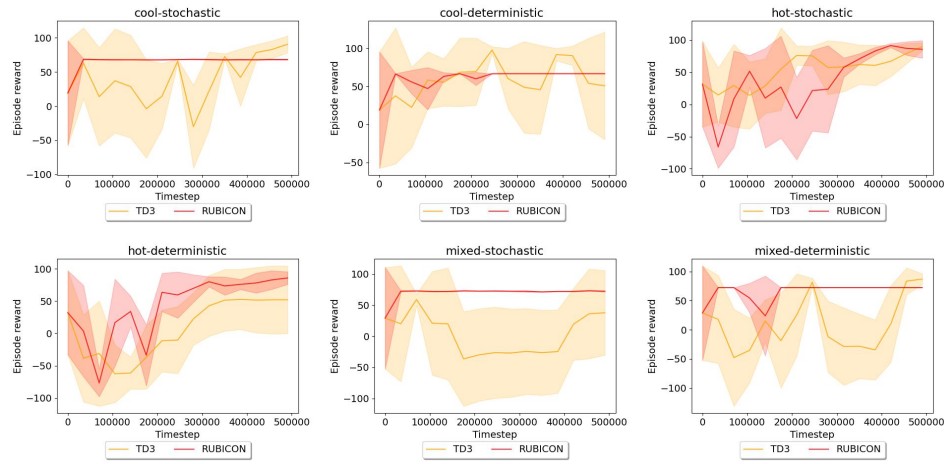

Figure 4: Learning curves comparing RUBICON and TD3, solid line indicates averaged values and half-transparent region is the range of one standard deviation.

in averaged scores and with a substantially smaller standard deviation across runs. The detailed learning curves are illustrated in Fig. 4.

| Environment | RUBICON | TD3 |
|---|---|---|
| hot-deterministic | **79.08±12.24** | 51.83±49.93 |
| mixed-deterministic | **72.34±0.00** | 23.46±41.01 |
| cool-deterministic | 66.52±0.00 | **66.3±41.66** |
| hot-stochastic | **83.64±8.25** | 71.8±21.51 |
| mixed-stochastic | **72.24±0.49** | 8.5±66.61 |
| cool-stochastic | 68.14±0.65 | **73.38±16.61** |
| Sum | **441.99±21.64** | 295.29±237.36 |

Table 5: RUBICON and TD3 comparison 1

| Environment | $\alpha = 1$ | $\alpha = 2.5$ | $\alpha = 4$ |
|---|---|---|---|
| hot-deterministic | **79.08±12.24** | 70.76±20.28 | 23.83±68.3 |
| mixed-deterministic | **72.34±0.00** | 71.92±0.52 | 72.26±0.05 |
| cool-deterministic | 66.52±0.00 | 53.28±23.81 | **68.16±4.69** |
| hot-stochastic | **83.64±8.25** | 73.92±20.24 | 65.13±28.08 |
| mixed-stochastic | **72.24±0.49** | **72.24±0.49** | **72.24±0.49** |
| cool-stochastic | **68.14±0.65** | 45.46±28.4 | 12.38±77.7 |
| Sum | **441.99±21.64** | 387.58±93.74 | 347.08±130.01 |

Table 6: Hyperparameter experiment. 1

### 5.2.1 HYPERPARAMETER EXPERIMENT

Since we integrate TD3 and TD3+BC as one, we need to introduce the weighting $\lambda$ and its hyperparameter $\alpha$ (see Eq. 2 and Alg. 1) in the online settings. It is mentioned in TD3+BC paper that the value of $\alpha$ decides if the model learns similar to RL ($\alpha=4$) or imitation learning ($\alpha=1$) and the default value set in TD3+BC is $\alpha=2.5$. We experiment on the values $\{1, 2.5, 4\}$ to observe how it affects the performance of our models in all tasks. The result (See Table 6) shows that when $\alpha=1$ the model gives the highest scores and the least variance. Which indicates that the actor policy should imitate RBC policy more in order to yield a more stable and robust averaged score rather than a traditional RL configuration ($\alpha=4$). In offline settings, we follow TD3+BC, use $\alpha=2.5$.

## 6 DISCUSSION

In this paper, we explore how rule-based control policy can be incorporated as regularization. Our method can be implemented on the baseline method with minimal changes. We apply our method in building HVAC control simulation environments in both online and offline settings. We empirically demonstrate that our method outperforms several state-of-the-art batch reinforcement learning methods and improves from its online baseline by a substantial amount in building HVAC tasks where rule-based control is robust and a standard in real-world settings. We expect our study would encourage both domain and RL experts to explore more opportunities for the combination of existing policies and RL and extend this concept to more real-world applications.

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

## A    EXPERIMENT DETAILS

**Software**

- **Python**: 3.9.12
- **Pytorch**: 1.12.1+cu113 (Paszke et al., 2019)
- **Sinergym**: 1.9.5 (Jiménez-Raboso et al., 2021)
- **Gym**: 0.21.0 (Brockman et al., 2016)
- **Numpy**: 1.23.1 (Van Der Walt et al., 2011)
- **CUDA**: 11.2

**Hardware**

- **CPU**: Intel Xeon Gold 6230 (2.10 GHz)
- **GPU**: NVidia RTX A6000

**Benchmark implementations**

- **DDPG**: We adopt the DDPG implementation in TD3 author-provided implementation
- **TD3**: Author-provided implementation
- **SAC**: We adopt CleanRL (Huang et al., 2021) implementation due to software version conflict with author-provided repository
- **TD3+BC**: author-provided implementation
- **CQL**: We adopt d3rlpy (Seno & Imai, 2021) implementation due to software version conflict with author-provided repository
- **BCQ**: Author-provided implementation

## B    BUILDING RL SETTINGS

In this section, we list all the details about the MDP settings in our problem.

- **State**: Site outdoor air dry bulb temperature, site outdoor air relative humidity, site wind speed, site wind direction, site diffuse solar radiation rate per area, site direct solar radiation rate per area, zone thermostat heating setpoint temperature, zone thermostat cooling setpoint temperature, zone air temperature, zone thermal comfort mean radiant temperature, zone air relative humidity, zone thermal comfort clothing value, zone thermal comfort Fanger model PPD (predicted percentage of dissatisfied), zone people occupant count, people air temperature, facility total HVAC electricity demand rate, current day, current month, and current hour.
- **Action**: Heating setpoint and cooling setpoint in continuous settings.
- **Reward**: We follow the default linear reward settings, it considers the energy consumption and the absolute difference to temperature comfort.
- **Environment**: A single floor building with an area of $463.6m^2$ divided in 5 zones, 1 interior and 4 exterior. The HVAC system is a packaged VAV (variable air volume) (DX (direct expansion) cooling coil and gas heating coils) with fully auto-sized input. The control variables are the cooling and heating temperature setpoints for the interior zone, and the simulation period is a full year (Jiménez-Raboso et al., 2021). The weather types are classified according to the U.S. Department of Energy (DOE) standard (Department of Energy). The weather type details and their representative geometric locations are listed below:
    - **Cool marine**: Washington, USA. The mean annual temperature and mean annual relative humidity are 9.3°C and 81.1% respectively.
    - **Hot dry**: Arizona, USA with mean annual temperature of 21.7°C and a mean annual relative humidity of 34.9%

- **Mixed humid**: New York, USA with a mean annual temperature of 12.6°C and a mean annual relative humidity of 68.5%

based on TMY3 datasets (National Renewable Energy Laboratory).

## C   LEARNING CURVES AND ADDITIONAL EXPERIMENTS

### C.1   LEARNING CURVES OF MAIN EXPERIMENTS

- **BRL learning** We illustrate the learning curves of BRL methods learn from different quality of buffers for a better visualization of comparison in Fig. 5 2 and 7
- **Transfer learning** The learning curves of BRL transfer experiments are illustrated in Fig. 8
- **Behavioral agents** The behavioral agents' learning curves are demonstrated in Fig. 9
- **Hyperparameter optimization** Learning curves of hyperparameter experiments in online RUBICON are shown in Fig. 10

### C.2   ADDITIONAL EXPERIMENTS

- **Learn from RBC buffers** To simulate a more realistic scenario where we learn from buffers of real-world HVAC control dataset. It consists of transitions generated from rule-based control policy, which is widely used in building HVAC control. The learning curves are illustrated in Figure 11. The results in Table 7 indicate that even when learning with RBC buffers with RBC policy itself, RUBICON could still outperforms RBC policy due to its learning ability.
- **CQL+RUBICON** Since CQL demonstrates a better performance compared to other methods except RUBICON (see Table 1), we conduct experiments combining CQL and our RUBICON method to learn from random buffers since we expect the most improvement in this scenario. The results in Figure 12 and Table 8 indicate that RUBICON also improves CQL. However, it does not consistently improve CQL's performance from task to task, which is not as we observe from TD3+BC to RUBICON learn from random buffers (see Figure 7). Also, the improvement is limited and not even reach the RBC policy performance, thus we did not continue exploring the possibility combining CQL with RUBICON.
- **Learn from a mixture of the original buffer and RBC buffer** In order to evaluate if mixing the buffers (of RBC buffer and the original buffer to learn from) is equivalent to RUBICON, we conduct experiments mixing 50% of transitions in RBC buffer with 50% of transition in the original buffer to learn from. The result is shown in Figure 14 and Table 10. It indicates our selective algorithm is necessary to dynamically decide if RBC policy or the behavioral policy to learn from instead of randomly trained on both.
- **Learn from worsened RBC policies** We run another ablation experiment to observe how the quality of RBC policy affects the performance compared with RUBICON and baseline TD3+BC. We design two worsened RBC policies: The first one is a biased RBC where we modify the change in setpoints ($a_{h_i}$ and $a_{c_i}$) from 1 to 5 in Algo. 2, we name this method "RBC_CB" in Figure 15. The other is to replace RBC with random policy, it is named as "RBC_Random". From the results in Table 11 we could find that even with constantly worsened RBC policy it still improves from baseline, However, it is still too aggressive for the models to learn a robust policy. And with random policy as a worsened RBC it is almost equivalent as no reference policy, the performance is similar to our baseline TD3+BC.

All learning curves are normalized with random policy as 0 and expert policy as 100, averaged with 3 random seeds and the scores shown in tables are the average and standard deviation last 5 evaluations unless mentioned otherwise.

## D   MODEL PARAMETERS

We list the hyperparameters used in this paper for reproducibility. Unless mentioned otherwise, we keep the original hyperparameters setups as the implementations listed in Sec. A since DRL methods are sensitive to hyperparameter tuning (Henderson et al., 2018) (see Table 12, 13, 14, and 15).

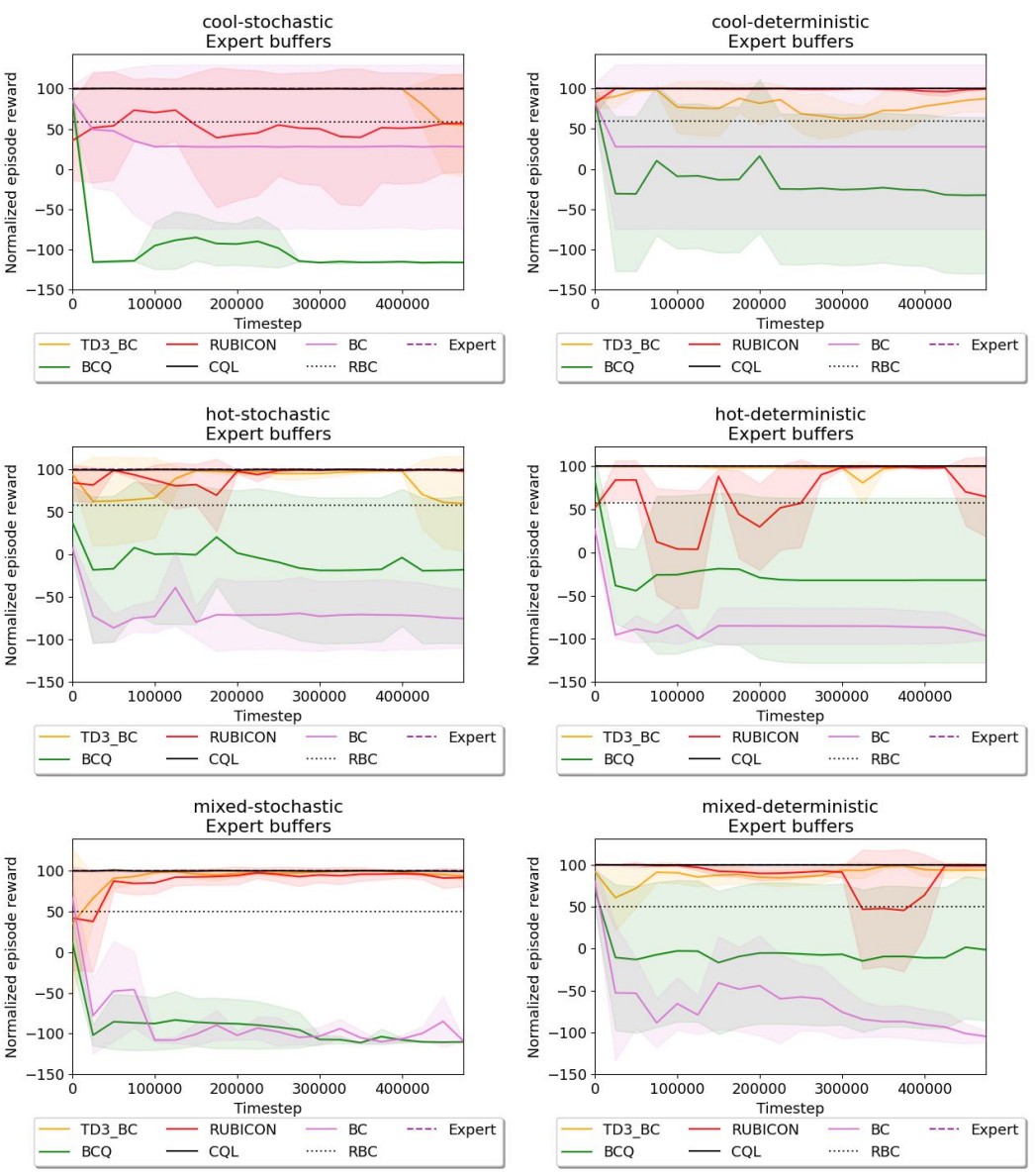

Figure 5: Learning curves of BRL models learn from expert buffers.

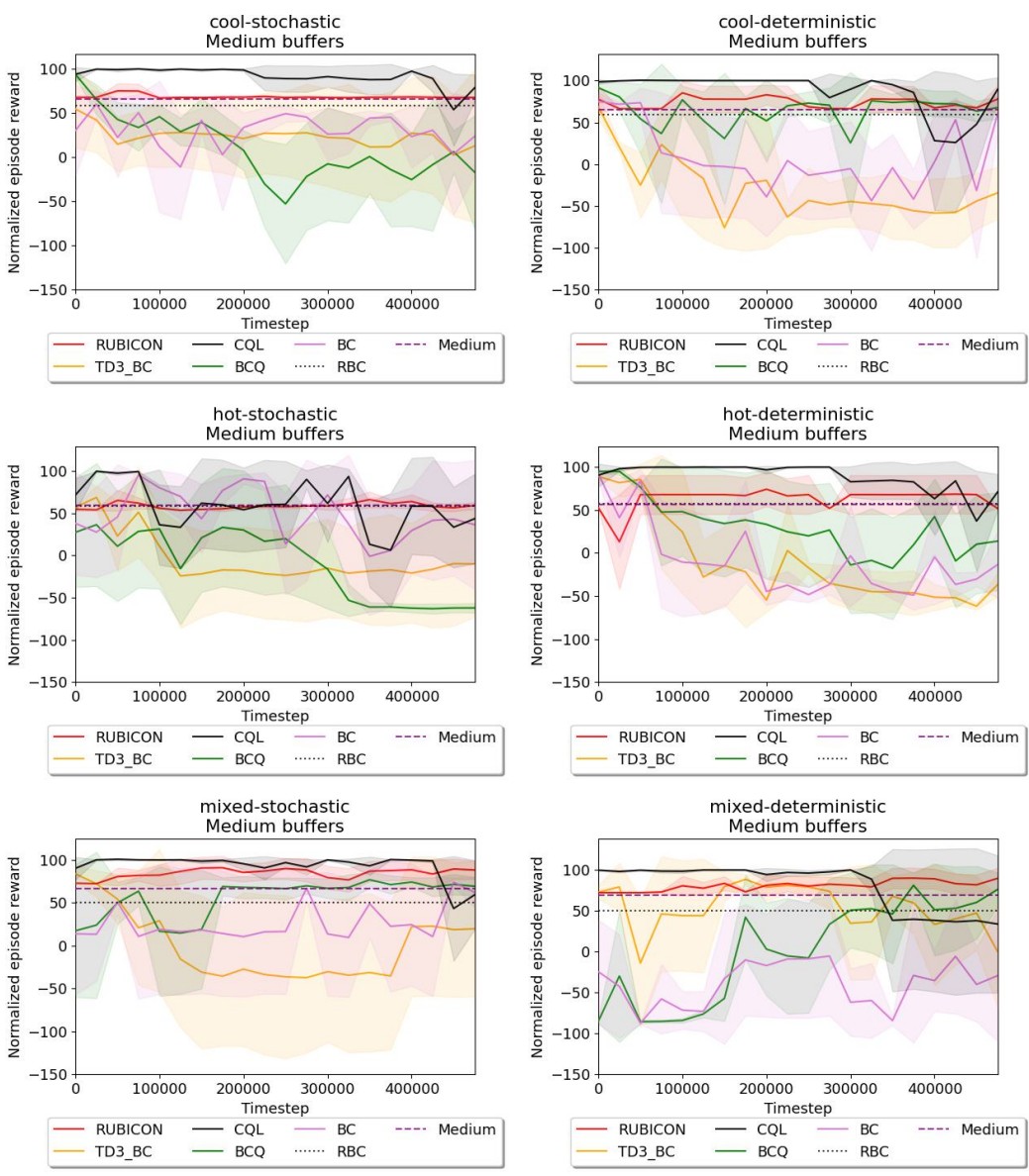

Figure 6: Learning curves of BRL models learn from medium buffers.

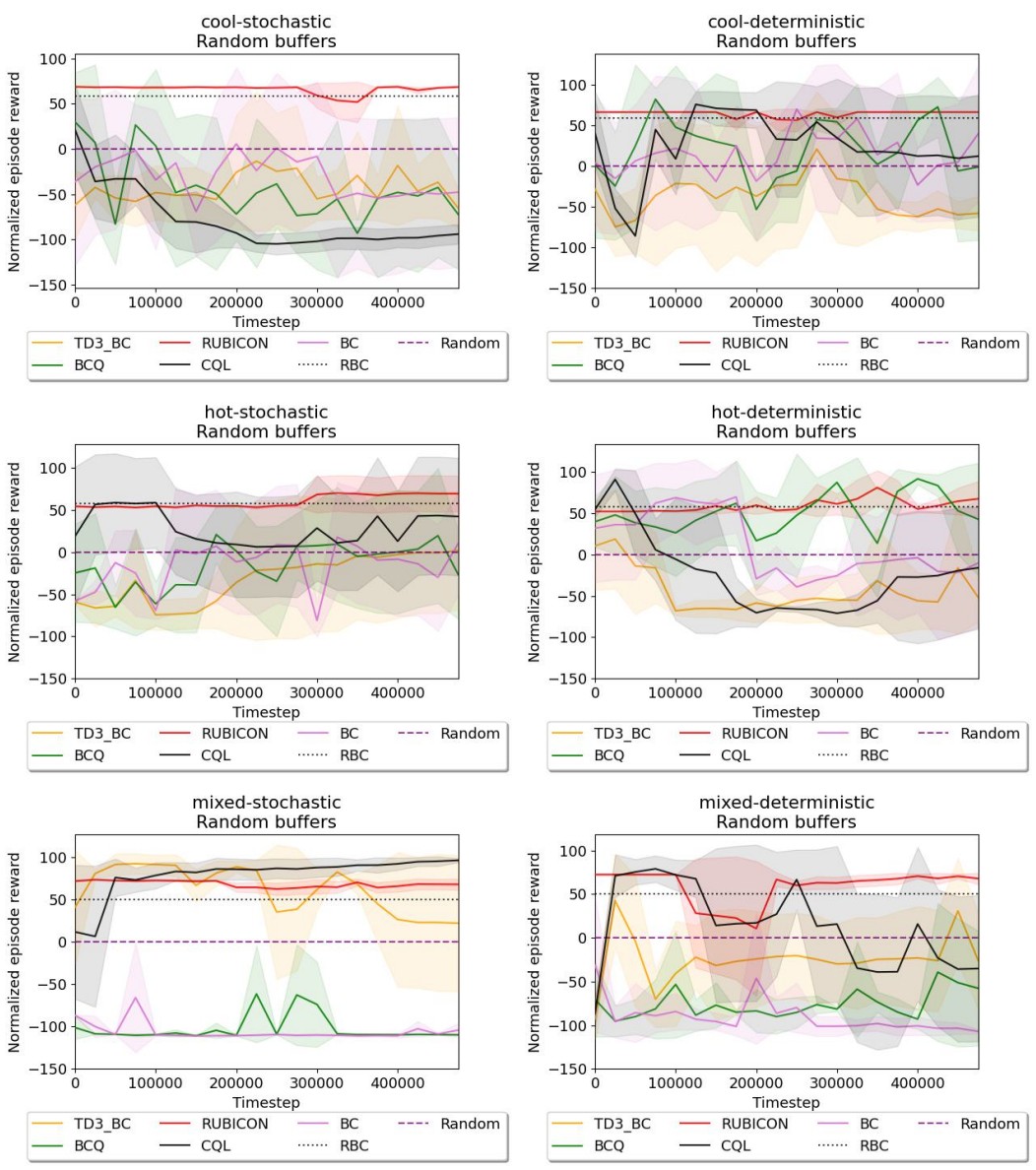

Figure 7: Learning curves of BRL models learn from random buffers

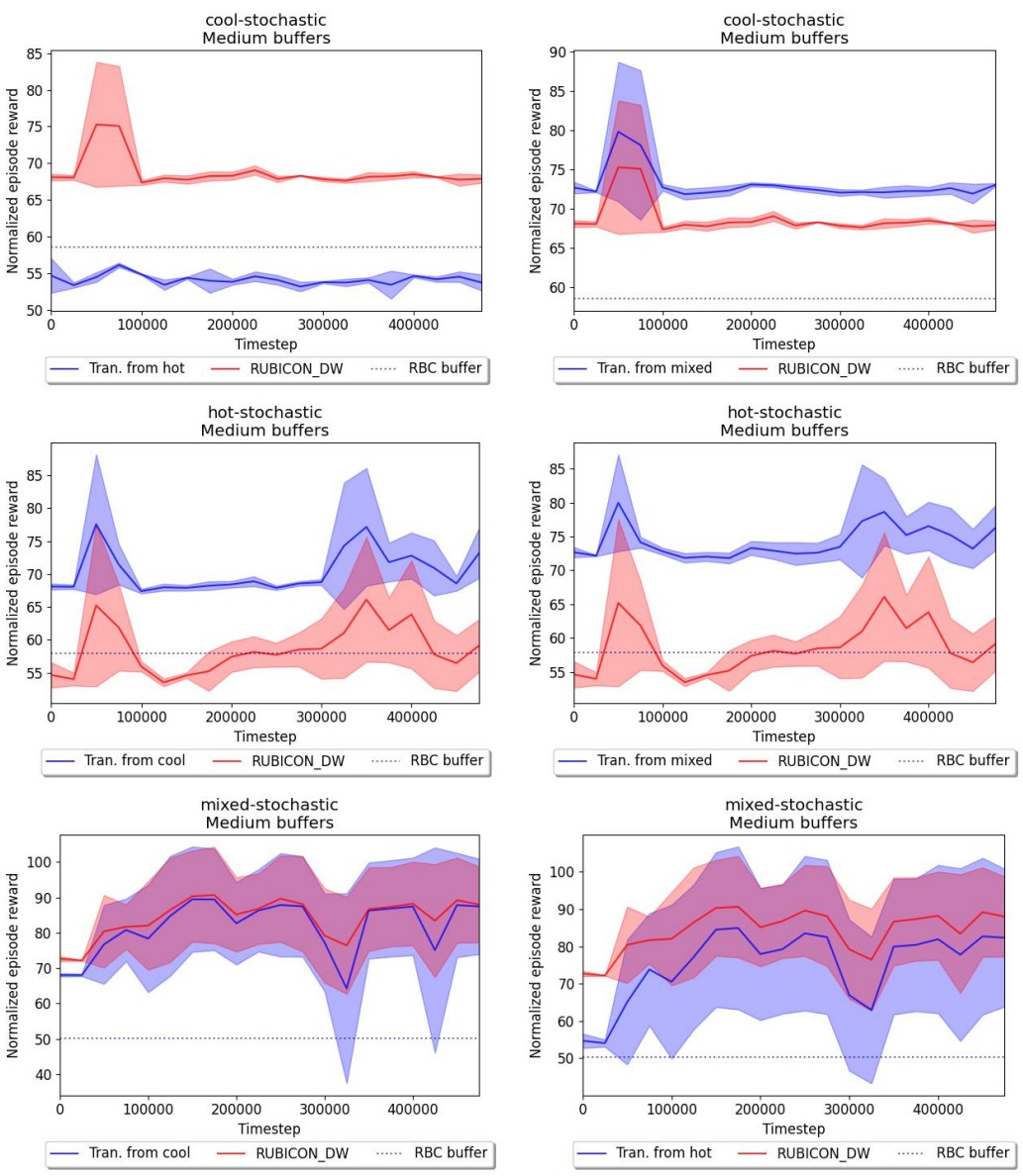

Figure 8: Learning curves of BRL models transferred from other weather types

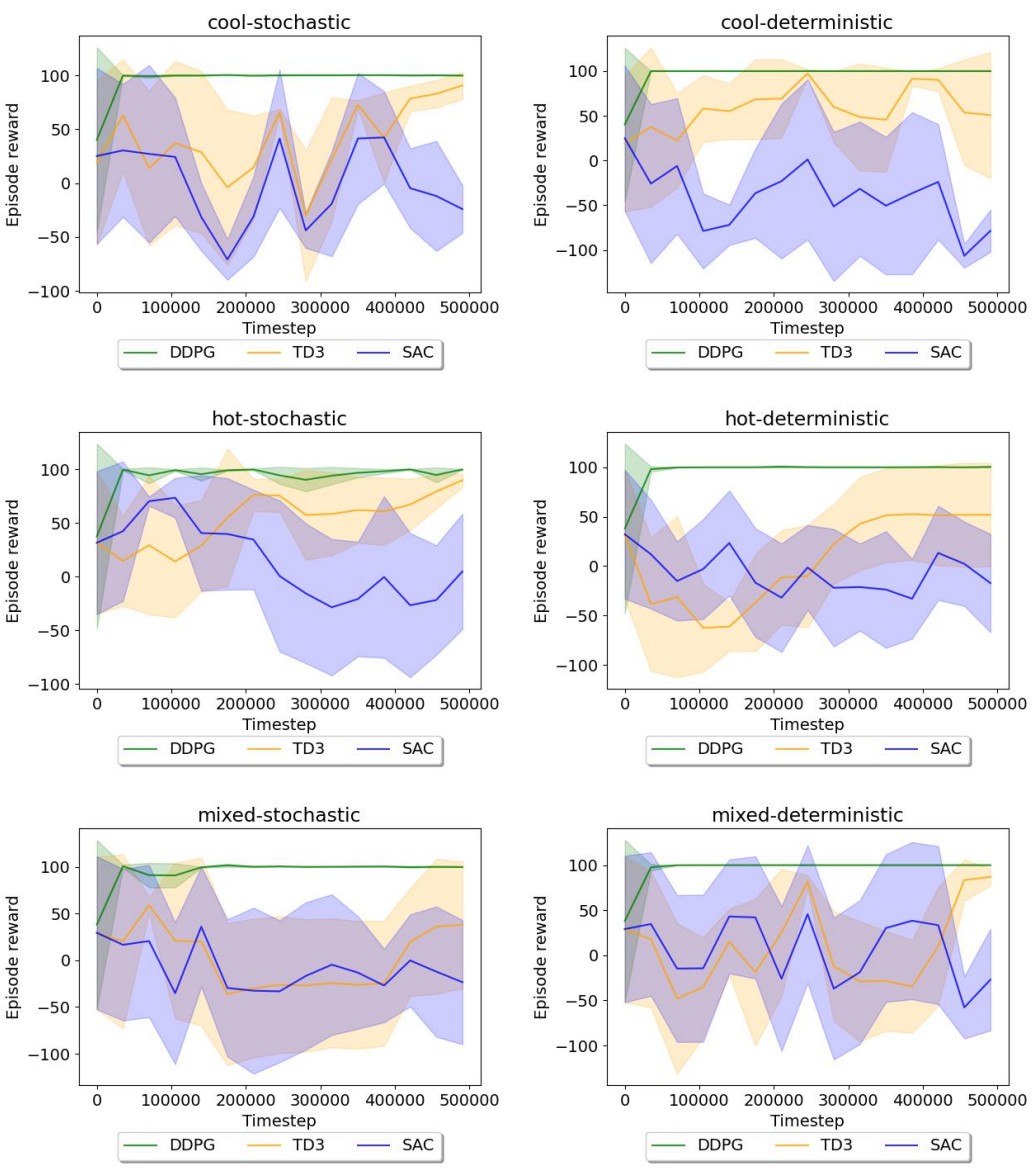

Figure 9: Learning curves of behavioral model training, behavioral models are trained with 500K time steps before generating buffers.

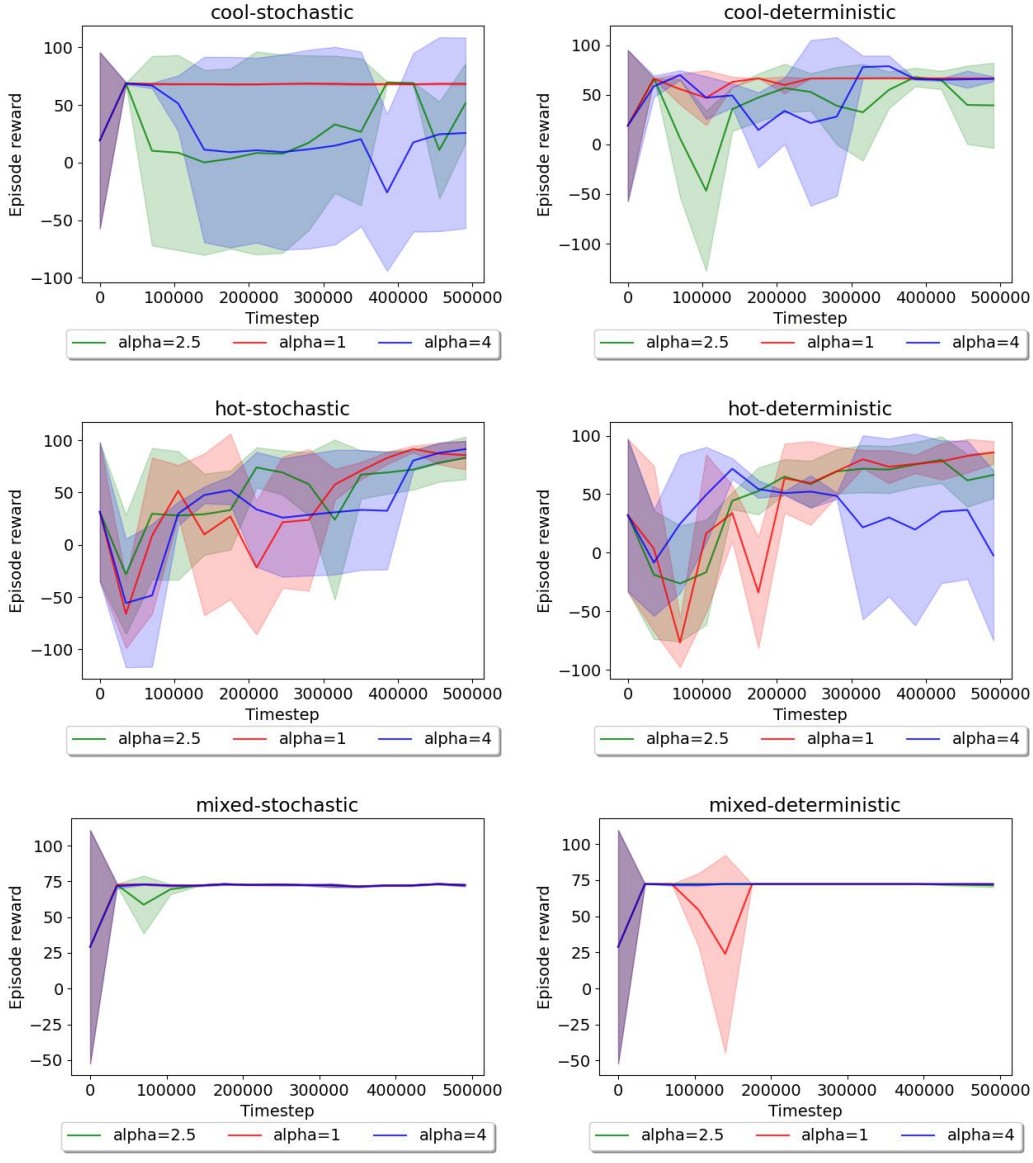

Figure 10: Learning curves of online RUBICON hyperparameter optimization.

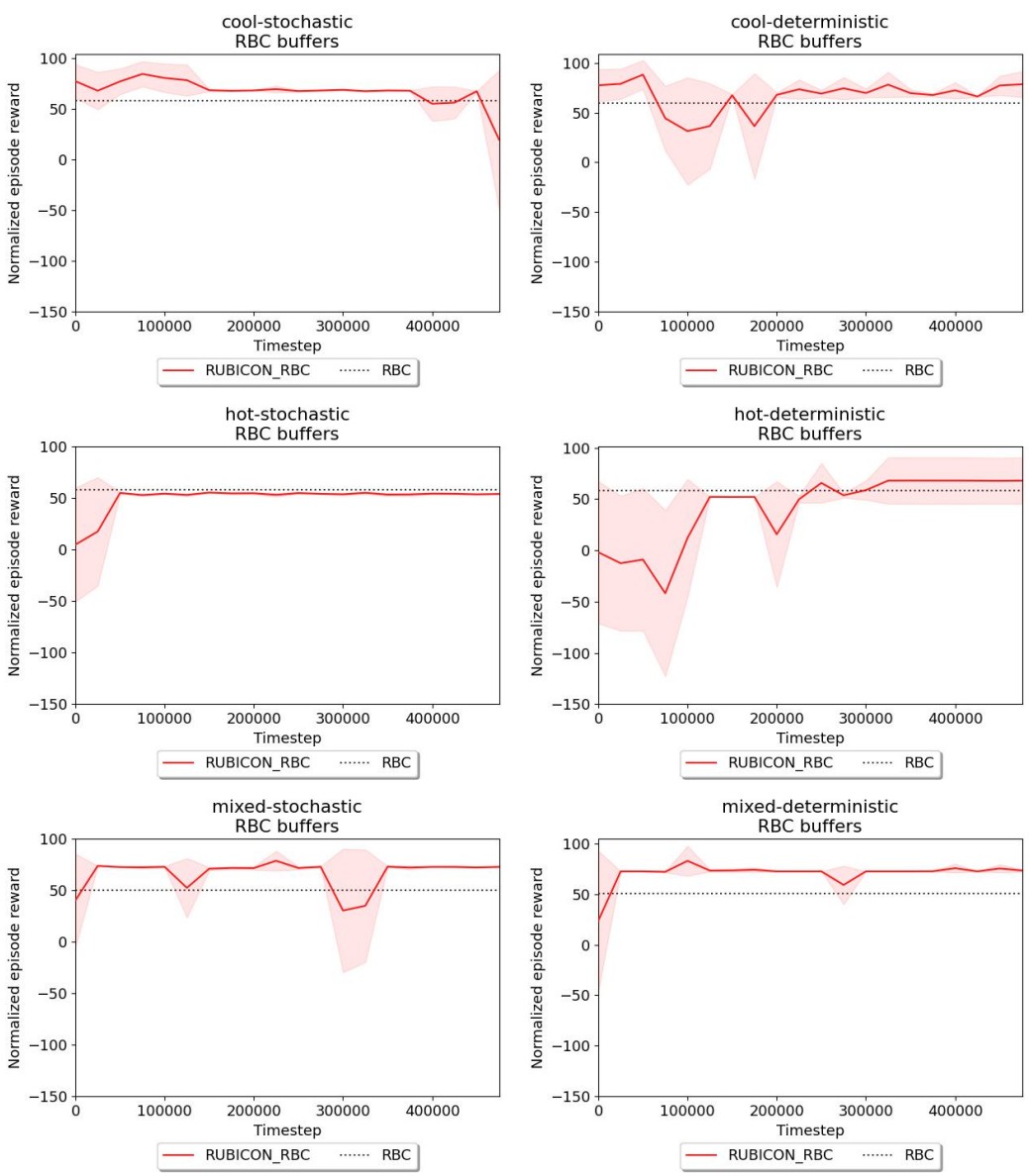

Figure 11: Learning curves of RUBICON learns from RBC buffers.

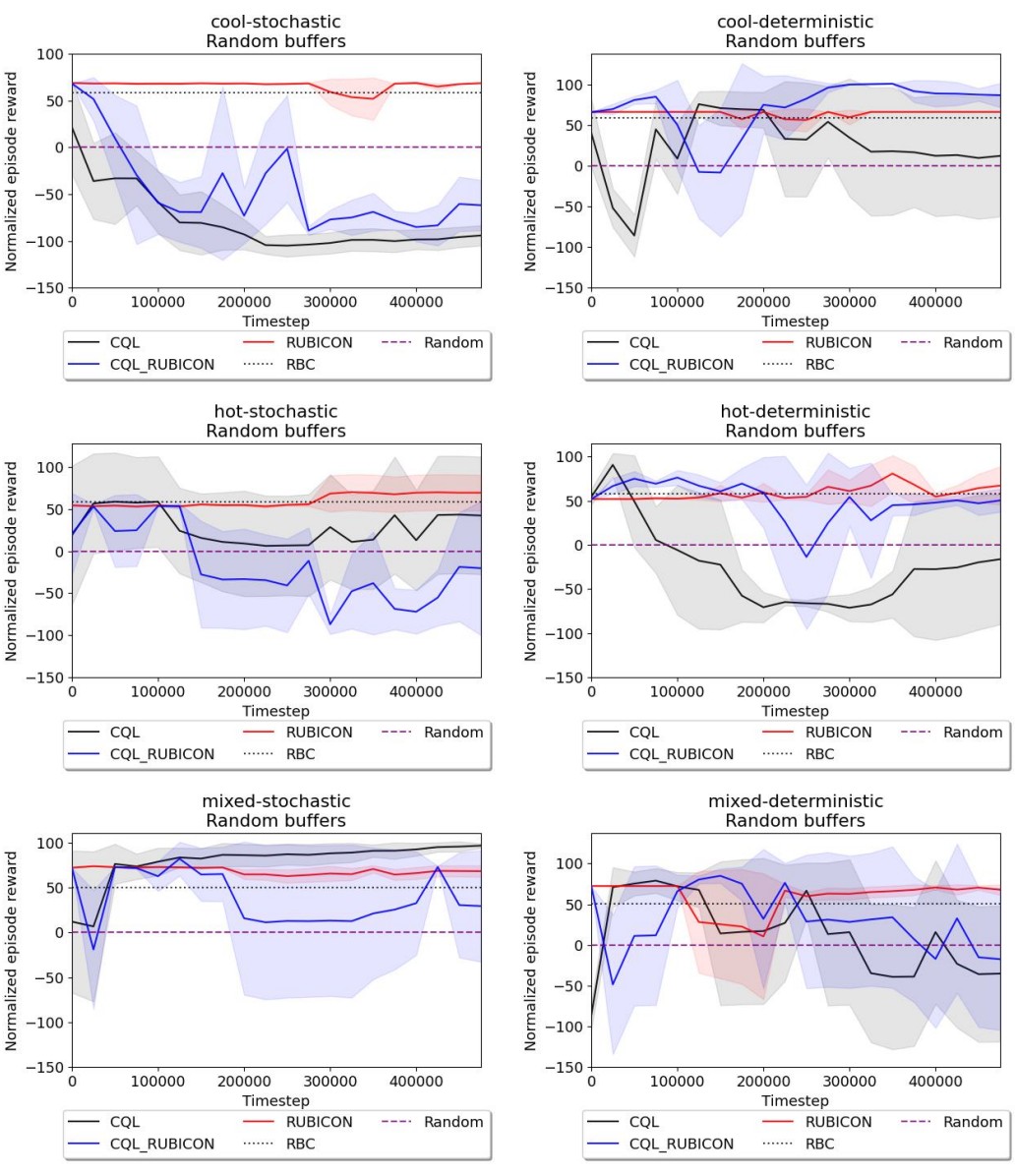

Figure 12: Learning curves of CQL, CQL+RUBICON, and RUBICON learn from random buffers

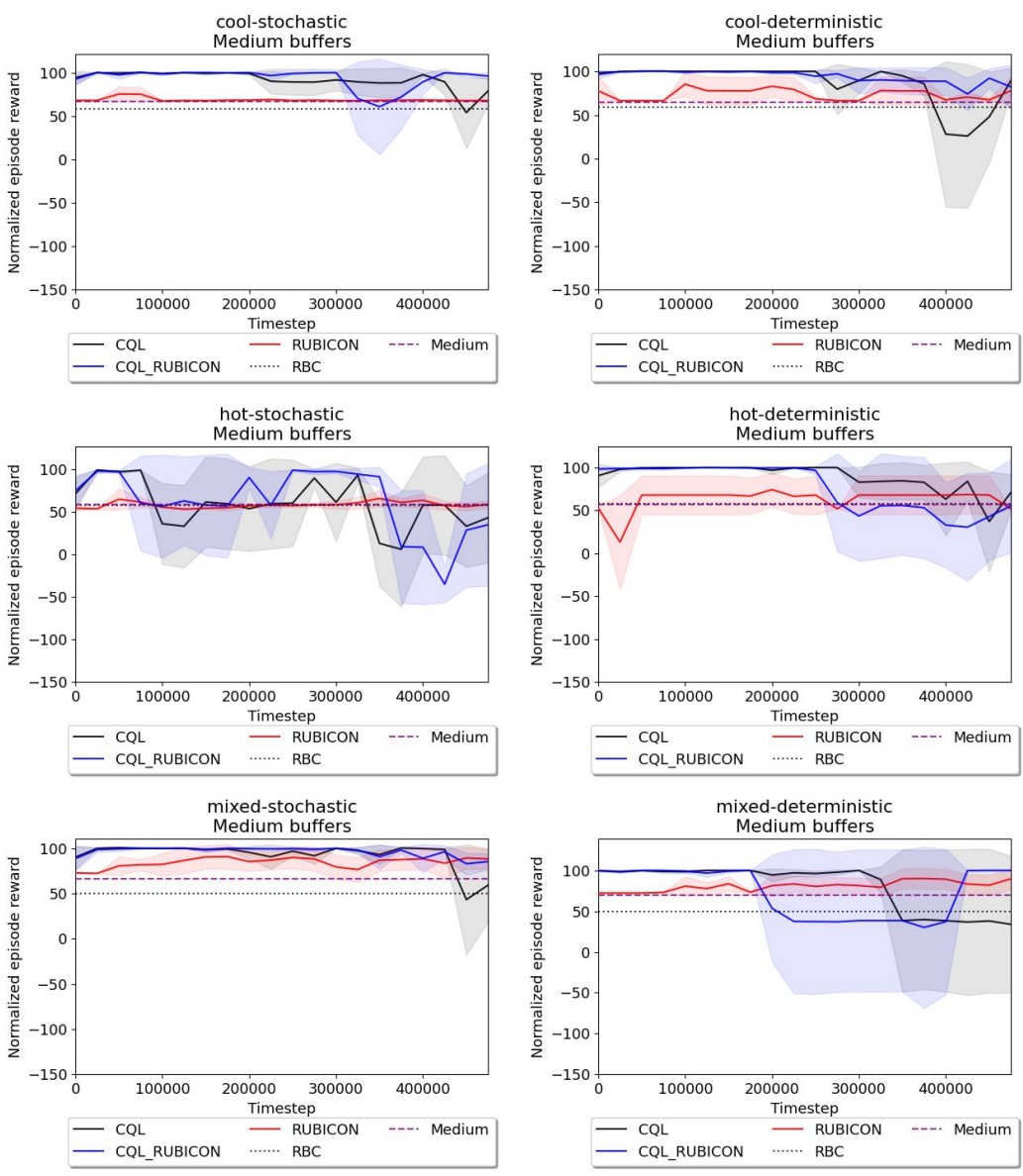

Figure 13: Learning curves of CQL, CQL+RUBICON, and RUBICON learn from medium buffers

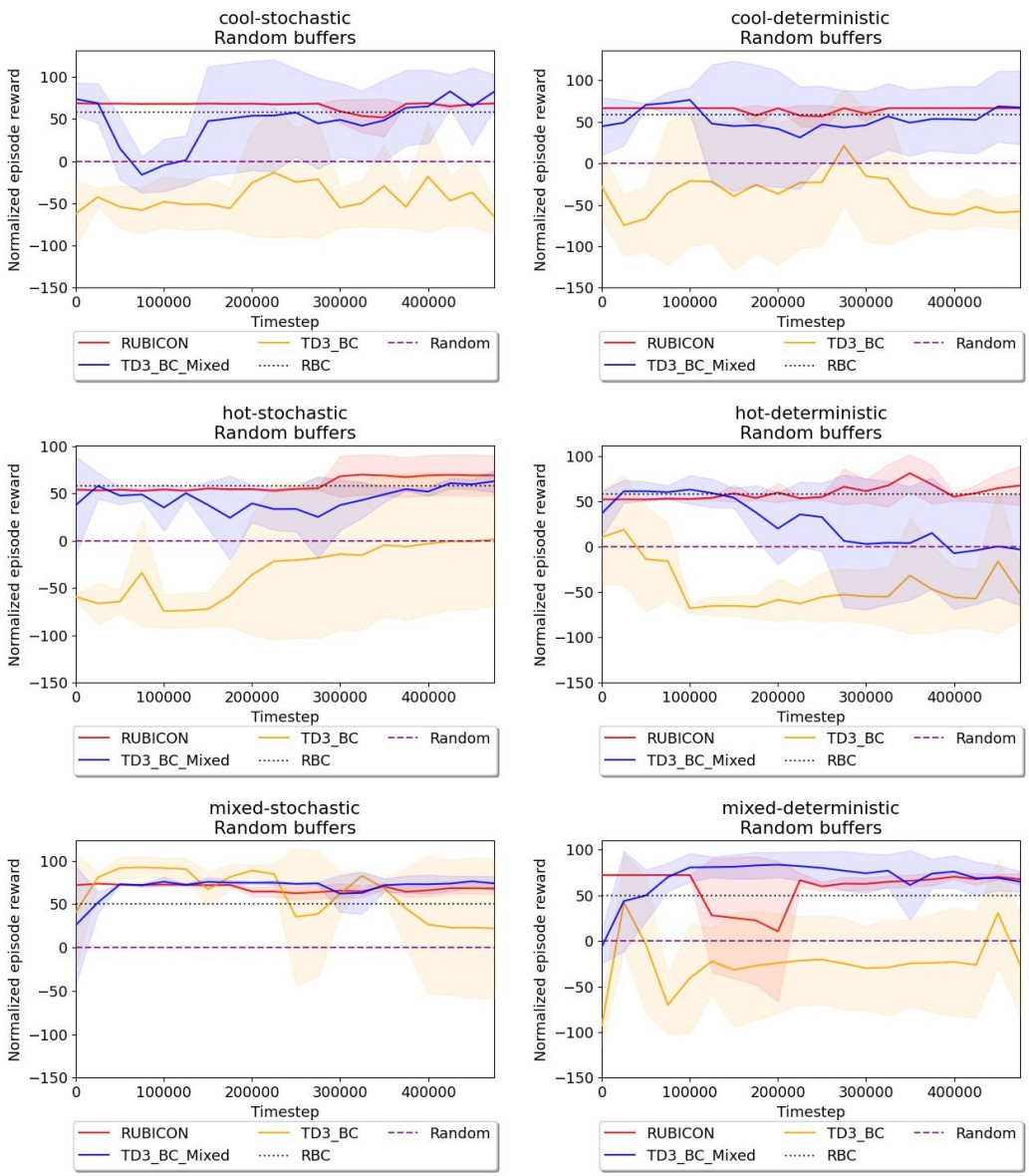

Figure 14: Learning curves comparing RUBICON and TD3+BC to TD3+BC learns from a mixture of 50% amount of transitions from the random buffer and 50% amount of transitions from the RBC buffer in stochastic environments

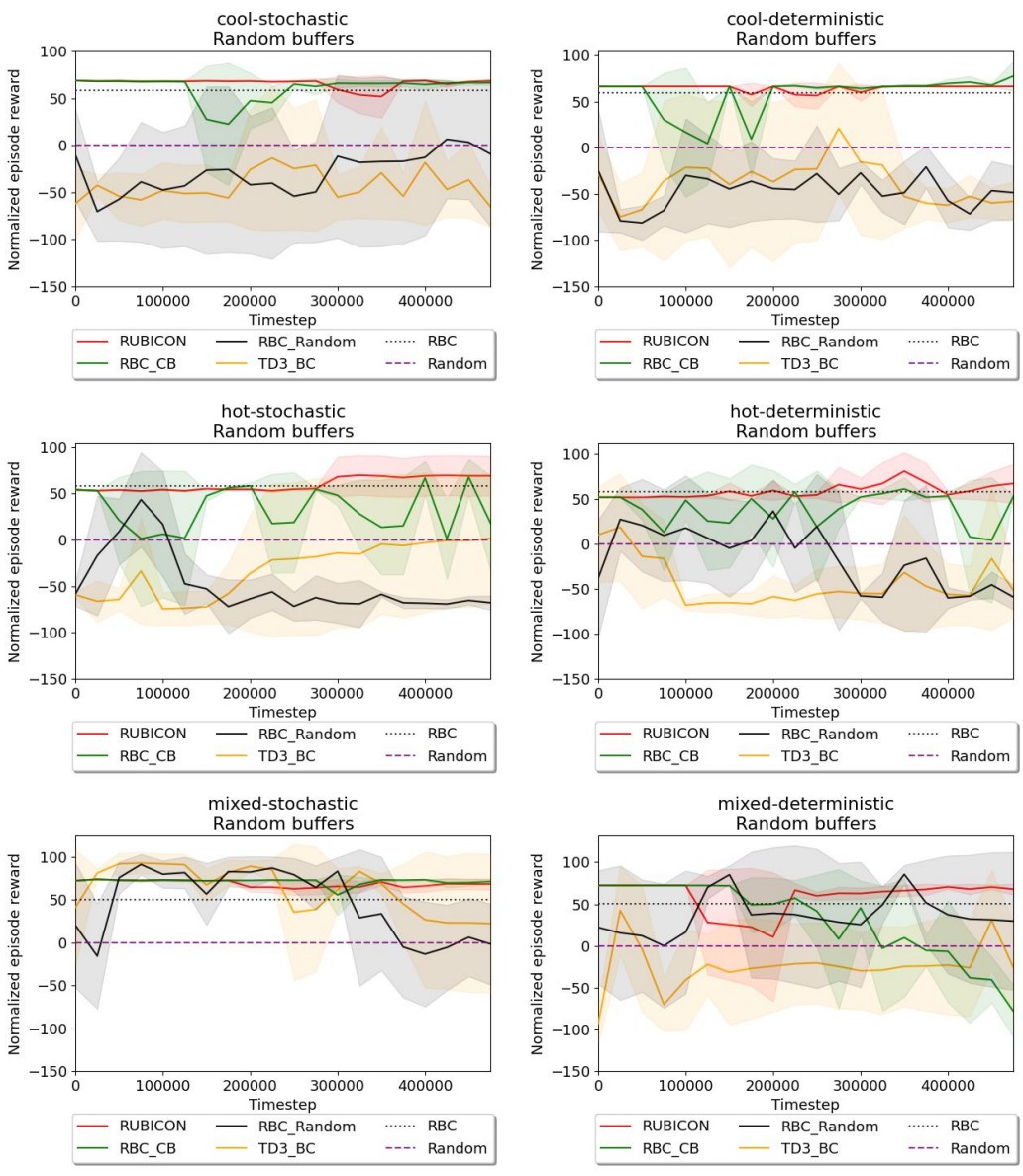

Figure 15: Learning curves of RUBICON learns from worsened RBC compared with TD3+BC and RUBICON

| Environment | RUBICON | RBC |
|---|---|---|
| hot-deterministic | **67.92±22.51** | 57.9 |
| mixed-deterministic | **73.68±1.96** | 50.12 |
| cool-deterministic | **72.28±6.53** | 59.15 |
| hot-stochastic | 53.83±0.8 | **57.92** |
| mixed-stochastic | **72.46±0.68** | 50.22 |
| cool-stochastic | 53.35±20.36 | **58.48** |
| Sum | **393.53±52.85** | 333.79 |

Table 7: RUBICON learns from buffers generated by RBC compared with RBC buffer performance

| Environment | RUBICON | CQL_RUBICON | CQL |
|---|---|---|---|
| hot-deterministic | **62.7±14.36** | 48.37±10.12 | -23.19±76.76 |
| mixed-deterministic | **68.83±4.93** | -2±85.18 | -23.46±83.61 |
| cool-deterministic | 66.5±0 | **88.98±13.24** | 12.98±73.04 |
| hot-stochastic | **68.83±21.26** | -47.04±45.48 | 36.64±67.61 |
| mixed-stochastic | 67.03±6.26 | 38.08±49.17 | **94.04±5.87** |
| cool-stochastic | **67.55±1.14** | -73.76±20.51 | -97.35±11.07 |
| Sum | **401.47±47.98** | 52.64±223.72 | -0.32±317.97 |

Table 8: CQL+RUBICON learns from random buffer compared with CQL and RUBICON 1

| Environment | RUBICON | CQL_RUBICON | CQL |
|---|---|---|---|
| hot-deterministic | 64.91±18.02 | 43.03±55.26 | **67.64±32.83** |
| mixed-deterministic | **86.84±12.39** | 73.4±37.59 | 37.36±86.8 |
| cool-deterministic | 72.2±8.07 | **85.24±17.33** | 55.44±49 |
| hot-stochastic | **59.72±5.29** | 9.39±58.72 | 39.92±56.67 |
| mixed-stochastic | 87.23±12.34 | **90.14±9.37** | 80.13±20.78 |
| cool-stochastic | 68.07±0.46 | **91.05±11.39** | 81.56±18.01 |
| Sum | **438.98±56.59** | 392.26±189.69 | 362.07±264.11 |

Table 9: CQL+RUBICON learns from medium buffer compared with CQL and RUBICON

| Environment | TD3+BC_Mixed | RUBICON | TD3+BC |
|---|---|---|---|
| hot-deterministic | 0.02±59.76 | **62.7±14.36** | -45.73±44.8 |
| mixed-deterministic | **70.66±15.45** | 68.83±4.93 | -13.71±57.06 |
| cool-deterministic | 59.01±40.92 | **66.5±0** | -58.4±19.25 |
| hot-stochastic | 57.93±5.6 | **68.83±21.26** | -1.82±73.31 |
| mixed-stochastic | **74.08±8.7** | 67.03±6.26 | 28.01±72.79 |
| cool-stochastic | **71.67±35.04** | 67.55±1.14 | -44.33±36.36 |
| Sum | 333.4±165.5 | **401.47±47.98** | -135.98±303.60 |

Table 10: TD3+BC learns from a mixture of random buffer and RBC buffer compared with RUBICON learns from random buffer 1

| Environment | RUBICON | TD3+BC | RBC_CB | RBC_Random |
|---|---|---|---|---|
| hot-deterministic | **62.7±14.36** | -45.73±44.8 | 34.06±27.13 | -47.72±25.11 |
| mixed-deterministic | **68.83±4.93** | -13.71±57.06 | -33.89±38.47 | 36.39±72.08 |
| cool-deterministic | 66.5±0 | -58.4±19.25 | **70.64±5.85** | -48.84±25.93 |
| hot-stochastic | **68.83±21.26** | -1.82±73.31 | 33.81±36.94 | -67.7±5.66 |
| mixed-stochastic | 67.03±6.26 | 28.01±72.79 | **71.22±2.64** | -4.07±52.46 |
| cool-stochastic | **67.55±1.14** | -44.33±36.36 | 65.84±3.06 | -5.9±74.12 |
| Sum | **401.47±47.98** | -135.98±303.60 | 241.69±114.12 | -137.85±255.38 |

Table 11: Comparison between RUBICON, TD3+BC, and worsened RBCs 1

| | Hyperparameter | Value |
|---|---|---|
| | Optimizer | Adam (Kingma & Ba, 2014) |
| | Critic learning rate | $3e^{-4}$ |
| | Actor learning rate | $3e^{-4}$ |
| | Mini-batch size | 256 |
| | Discount factor | 0.99 |
| Algorithm hyperparameters | Target update rate | $5e^{-3}$ |
| | Policy noise | 0.2 |
| | Policy noise clipping | (-0.5, 0.5) |
| | Policy update frequency | 2 |
| | TD3+BC $\alpha$ | 2.5 |
| | RUBICON online $\alpha$ | 1 |
| | RUBICON offline $\alpha$ | 2.5 |
| | RUBICON online $\xi$ | 0 if $\bar{Q}_{\mathcal{B}}(s, \pi_b(s)) \geq \bar{Q}_{\mathcal{B}}(s, \pi_{rbc}(s))$ else 1 |
| | RUBICON offline $\xi$ | 1 |
| | Critic hidden dimension | 256 |
| | Critic hidden layers | 2 |
| Network architecture | Critic activation function | ReLU |
| | Actor hidden dimension | 256 |
| | Actor hidden layers | 2 |
| | Actor activation function | ReLU |

Table 12: TD3, TD3+BC, and RUBICON hyperparameters

| | Hyperparameter | Value |
|---|---|---|
| | Optimizer | Adam |
| | Critic learning rate | $1e^{-3}$ |
| | Actor learning rate | $3e^{-4}/1e^{-4}$ |
| | Mini-batch size | 256 |
| | Discount factor | 0.99 |
| | Target update rate | $5e^{-3}$ |
| Algorithm hyperparameters | Policy noise | 0.2 |
| | Policy noise clipping | (-0.5, 0.5) |
| | Policy update frequency | 2 |
| | SAC entropy auto-tuning | True |
| | CQL $\alpha$ threshold | 10 |
| | CQL conservative weight | 5.0 |
| | CQL number of sampled actions | 10 |
| | Critic hidden dimension | 256 |
| | Critic hidden layers | 3 |
| Network architecture | Critic activation function | ReLU |
| | Actor hidden dimension | 256 |
| | Actor hidden layers | 3 |
| | Actor activation function | ReLU |

Table 13: SAC/CQL hyperparameters

|  | Hyperparameter | Value |
|---|---|---|
|  | Optimizer | Adam |
|  | Critic learning rate | $1e^{-3}$ |
|  | Actor learning rate | $1e^{-4}$ |
|  | Mini-batch size | 64 |
| Algorithm hyperparameters | Discount factor | 0.99 |
|  | Target update rate | $1e^{-3}$ |
|  | Policy noise | $\mathcal{N}(0, 0.1)$ |
|  | Policy noise clipping | (-0.5, 0.5) |
|  | Policy update frequency | 1 |
|  | Critic hidden dimension | 400/300 |
|  | Critic hidden layers | 2 |
|  | Critic activation function | ReLU |
| Network architecture | Actor hidden dimension | 400/300 |
|  | Actor hidden layers | 2 |
|  | Actor activation function | ReLU |

Table 14: DDPG hyperparameters

|  | Hyperparameter | Value |
|---|---|---|
|  | Optimizer | Adam |
|  | Critic learning rate | $1e^{-3}$ |
|  | Actor learning rate | $1e^{-4}$ |
|  | Mini-batch size | 100 |
| Algorithm hyperparameters | Discount factor | 0.99 |
|  | Target update rate | $5e^{-3}$ |
|  | Minimum weighting | 0.75 |
|  | Max perturbation | 0.05 |
|  | Critic hidden dimension | 400/300 |
|  | Critic hidden layers | 2 |
|  | Critic activation function | ReLU |
|  | Actor hidden dimension | 400/300 |
| Network architecture | Actor hidden layers | 2 |
|  | Actor activation function | ReLU |
|  | VAE hidden dimension | 750 |
|  | VAE latent vector clipping | (-0.5, 0.5) |

Table 15: BCQ/BC hyperparameters

