# OpenReview forum: "Rule-based policy regularization for reinforcement learning-based building control"
_ICLR.cc/2023/Conference — Submitted to ICLR 2023_

### Official Review · Reviewer_8sr1 · 2022-10-18

**Confidence:** 5
**Clarity, Quality, Novelty And Reproducibility:** 1. Table 1 shows that RUBICON perform…
**Correctness:** 2
**Technical Novelty And Significance:** 1
**Empirical Novelty And Significance:** 1
**Recommendation:** 3

**Strength And Weaknesses:**

Strength
1. The proposed method utilizes the rule-based control policy to improve stability in offline and online RL settings.
2. The experimental results show that the proposed method performs better than the baseline methods in offline and online RL settings.

Weakness
1. There is no theoretical justification.
2. The experimental results of the offline setting show that the performance of the proposed method is competitive with CQL although CQL does not exploit the rule-based controller.
3. The novelty of RUBICON in the online RL setting is unclear. For example, it should be compared with Residual Policy Learning.


**Summary Of The Paper:**

This paper proposes a method that integrates a rule-based control policy with online/offline reinforcement learning. The proposed method, RUBICON, is an extension of TD3+BC, which is one of the state-of-the-art offline RL algorithms. The basic idea is to switch the actor's loss function based on the state-action value function. The proposed method is evaluated on the HVAC control task. RUBICON outperformed the baseline methods in offline and online RL settings.

**Summary Of The Review:**

Although the proposed method performed better than the baseline methods, I think the comparison is unfair because the baseline methods do not use the knowledge/data of the rule-based controller. In the offline setting, CQL performed better when the expert agent collected the training data. In the online setting, the proposed method should be compared with Residual Policy Learning and other baselines. Thus, I think this paper is not ready for publication.

---

> ### Author Response · Authors · 2022-11-19
> **Responses to reviewer**
>
> We appreciate the valuable review, our responses are listed below.
>
> - **There is no theoretical justification.**
>   - Our contributions are based on empirical results. Such papers are common in ICLR [1, 2]. Our method outperforms state-of-the-art methods in building HVAC control tasks.
>   [1] Zhang, X. and Wu, D.., "Empirical Studies on the Properties of Linear Regions in Deep Neural Networks.", ICLR 2019.
>   [2] Toneva, M., Sordoni, A., des Combes, R.T., Trischler, A., Bengio, Y. and Gordon, G.J. "An Empirical Study of Example Forgetting during Deep Neural Network Learning.", ICLR 2018.
>   [3] Andrychowicz, M., Wolski, F., Ray, A., Schneider, J., Fong, R., Welinder, P. & Zaremba, W., "Hindsight experience replay.", NeurIPS 2017.
>
> - **The experimental results of the offline setting show that the performance of the proposed method is competitive with CQL although CQL does not exploit the rule-based controller.**
>   - Our focus is on improving a baseline algorithm by integrating an existing rule-based policy. To extend our analysis, we ran experiments with CQL+RUBICON, where we integrate RBC policy with CQL instead. The experimental results (Figure 12, 13 and Table 8, 9) show that our method does improve on CQL. Detailed description is in Appendix C.2. However, TD3BC+RUBICON performs the best overall.
>
> - **Table 1 shows that RUBICON performed worse than CQL when the training data was collected by an expert agent. It suggests that dynamic switching of the actor's loss based on the value function does not work correctly.**
>   - RUBICON underperforms the baseline algorithms (CQL and TD3+BC) when the data is from an expert buffer. We hypothesize that the Q-value estimates of the actions taken by rule-based policy have estimation errors in this case, and impacts performance. We will add this as a limitation in the paper.
>
> - **When the data collected by the rule-based controller is added to the buffer, do the previous offline RL methods improve performance?**
>   - We have added new experiments that use a mix (50%, randomly selected) of RBC data into the existing buffers. We use TD3+BC as a representative offline RL method. We include this experiment in Appendix C.2 (Fig. 14 and Table 10). Our algorithm outperforms this version of TD3+BC.
>
> - **The novelty of RUBICON in the online RL setting is unclear. For example, it should be compared with Residual Policy Learning.
> The proposed method should be compared with residual policy learning and TD3 with demonstration in the online setting. In particular, the latter also adds the behavior cloning loss function to the RL objective. Silver et al. (2018). Residual Policy Learning. arXiv. He et al. (2020). Deep Reinforcement Learning based Local Planner for UAV Obstacle Avoidance using Demonstration Data. arXiv.**
>
>   - Thank you for the suggestion. The papers cited are not published in any peer-reviewed venues. Following [ICLR guidelines](https://iclr.cc/Conferences/2023/ReviewerGuide), we do not include them in related work or evaluation.
>   - Silver et al. (2018) use demonstrations, and is similar in spirit to our problem statement. However, their algorithm requires specification of goal states, similar to Hindsight Experience Replay. In building control systems, such goal states are not defined, instead we optimize for energy use over a horizon. We cannot use Residual Policy Learning or Hindsight Experience Replay for our problem.
>   - He et al. (2020) use demonstrations to guide online reinforcement learning. The algorithm is similar to TD3+BC, which is a state-of-the-art algorithm in this domain. We compare our RUBICON algorithm against TD3+BC in the paper.
>
> - **The titles of Figures 2 and 4 should be modified. For example, Eplus-5Zone-cool-continuous-stochastic-v1 is not understandable. I think it means the "cool-continuous" environment. Is it correct?**
>   - Yes, “Eplus-5Zone-cool-continuous-stochastic-v1” means "cool-continuous". We have modified all figures with consistent naming to avoid confusion.
>
> - **Although the proposed method performed better than the baseline methods, I think the comparison is unfair because the baseline methods do not use the knowledge/data of the rule-based controller. In the offline setting, CQL performed better when the expert agent collected the training data. In the online setting, the proposed method should be compared with Residual Policy Learning and other baselines. Thus, I think this paper is not ready for publication.**
>   - The focus of our paper is to improve algorithm performance in the building controls application. Our algorithm outperforms state-of-the-art in this domain. To our knowledge, we are the first to integrate rule-based policy algorithmically in existing offline and online RL methods. We do not seek to make fair comparisons against CQL (which does not make use of rule-based policy) or Residual Policy Learning (which is not applicable to our application setting, and is not peer-reviewed).

---

### Official Review · Reviewer_r7Hj · 2022-10-25

**Confidence:** 4
**Correctness:** 3
**Technical Novelty And Significance:** 3
**Empirical Novelty And Significance:** 3
**Recommendation:** 6

**Clarity, Quality, Novelty And Reproducibility:**

Clarity: method is clear to understand, but the results and the presentation of results could be improved

quality and novelty: The method demonstrates good performance and the combination of rule-based control with reinforcement learning makes a lot of sense and potentially has huge impacts. the experiments are strong to demonstrate and validate the characteristics of the proposed method (its pros and cons)

reproducibility: code is not provided, so I am not sure if it could be reproduced.

**Strength And Weaknesses:**

Strength: the paper is well written and contributions are appropriately stated, and the algorithm itself is easy to understand. The performed experiments show that the proposed algorithm is useful in some cases (medium and random buffer, etc.)

Weakness: (1) the figures are not designed carefully. For example, figure 1, it is not clear what each arrow mean, and it seems "solid lines" not always means "fixed actions". figure 2, the difference between different methods aren't that huge to distinguish one from another. Figure 3, it is not clear why this "RUBICON selects actions in a wider range" and "TD3+BC has a reward distribution of higher values", since it's hard to visually validate this, and better using quantitative approach if the author really want to prove this.

(2) some results are mixed. sometimes the proposed algorithm performs better and sometimes not, the provided explanations are not very convincing. For example, why in hot-stochastic case, RUBICON performs better in table 1 with. expert data, while other cases, with expert data, RUBICON performs worse. Some results have 0.0 standard deviation, is that correct or error in data recording? is there a reason why it is exactly zero?


**Summary Of The Paper:**

This work proposes a method to unify rule-based control with reinforcement learning for building management and control. The work provides both the online and offline setting variants. The idea is to extend the previous work of TD3 + BC by replacing the behavior policy with a dynamically weighted policy that chooses action either from the behavior policy or the rule-based policy. In TD3+BC, the policy update rule is updated to both maximize the value w.r.t. the policy, and also minimize the difference of the policy with a behavior cloning policy. In order to incorporate rule-based control, the method proposes to choose whichever policy with the maximum value from the collection of behavior cloning policy and rule-based policy. Experiments are performed in the offline setting and online setting. The offline setting demonstrates that the method has better data efficiency than other baselines, and is able to transfer from one weather condition to another with comparable performance. The online experiment also demonstrates that the method is better than TD3.

**Summary Of The Review:**

Overall, I think the method is clear and strong, and results are good but could be improved. If the author could provide a reproducible work that would be even better.

---

> ### Author Response · Authors · 2022-11-18
> **Responses to reviewer**
>
> We appreciate the valuable review, our responses are listed below.
>
> - **(1) the figures are not designed carefully.**
>   - Thank you for the suggestions. We have a number of edits in our figures to improve readability and clarity.
>
> - **For example, figure 1, it is not clear what each arrow mean, and it seems "solid lines" not always means "fixed actions".**
>   - We have modified Figure 1 to add more detailed descriptions and use solid lines everywhere to avoid confusion.
>
> - **figure 2, the difference between different methods aren't that huge to distinguish one from another.**
>   - We have modified the transparency of the range within ±1 standard deviation for a better visualization.
>
> - **Figure 3, it is not clear why this "RUBICON selects actions in a wider range" and "TD3+BC has a reward distribution of higher values", since it's hard to visually validate this, and better using quantitative approach if the author really want to prove this.**
>   - We have added the values of a1 range, a2 range, and reward mean in the caption of Figure 3.
>
> -- **(2) some results are mixed. sometimes the proposed algorithm performs better and sometimes not, the provided explanations are not very convincing. For example, why in hot-stochastic case, RUBICON performs better in table 1 with. expert data, while in other cases, with expert data, RUBICON performs worse.**
>   - In expert buffers, the action range is narrower, and thus the action-value estimation is less accurate than other buffers when there is unseen action-value pair from the RBC policy. This leads to worse performance of RUBICON learning with expert buffers. We will add this as a limitation in the paper.
>   - To validate this assumption, we list the average action range (averaged across tasks) of buffers in the table below:
> | Range/Buffer | Expert | Medium | Random |  RBC  |
> |:------------:|:------:|:------:|:------:|:-----:|
> |   a1-range   |  0.482 |  0.906 |  1.999 | 0.555 |
> |   a2-range   |  0.459 |  0.952 |  1.999 | 0.741 |
>
> - **Some results have 0.0 standard deviation, is that correct or error in data recording?**
>   - Thanks for pointing this out. We have corrected the standard deviation values in all the tables, and fixed the bug of calculating the standard deviation and the rounding error.
> - **is there a reason why it is exactly zero?**
>   - Since we round down the decimal points to three digits in normalized scores, the extremely small numbers are ignored in these calculations. We have added footnotes for tables with any standard deviation of 0.
>
> - **reproducibility: code is not provided, so I am not sure if it could be reproduced.**
> We will open-source our code, baselines, and data.

---

### Official Review · Reviewer_6eMZ · 2022-10-30

**Confidence:** 4
**Correctness:** 3
**Technical Novelty And Significance:** 3
**Empirical Novelty And Significance:** 3
**Recommendation:** 5

**Clarity, Quality, Novelty And Reproducibility:**

The paper is clearly written. Certainly the novelty in the idea is low, but I encourage authors to reformulate the wording of the paper to be more of an applications paper -- Applying offline RL for building control, and provide benchmarks for RL researchers to iterate on. I think that could be a big win of this paper if done right.

**Strength And Weaknesses:**

Strength: While the idea of combining rule based learning with RL is not new, the paper presents an example with offline RL with a convincing downstream application (building control).

Weaknesses:

- Can the rule based policy also be learned, like parameters of a rule, to make it robust? More generally, how robust is this approach if the building changes? Can we see an ablation study where we compare performance of the method to standard offline RL when the rule based policy becomes worse and worse?

- Since TD3+BC is worse than CQL, as evident from the table, can this method also be applied to CQL to see how well it performs?

- Would be good to do some qualitative analysis: when does the RL policy perform particularly much better than the rule-based policy? What happens if the data composition is different (more representative of real-world systems than random, medium, expert from D4RL)?

I like this paper overall, and would encourage authors to answer the questions above, after which I am happy to increase my score.

**Summary Of The Paper:**

This paper presents an approach for combining rule based policies and RL methods, such as TD3+BC and TD3. The proposed approach performs well (though I have no clear calibration of the tasks being benchmarked upon). The resulting policy attains good performance and can improve upon heuristics obtained through the rule based policy.

**Summary Of The Review:**

Overall, I like the paper, but I am giving a reject score right now because I think the paper would be much stronger if my weaknesses are addressed. if the authors can address those weaknesses, I will move up to an accept.

---

> ### Author Response · Authors · 2022-11-18
> **Responses to reviewer**
>
> We appreciate the valuable review, our responses are listed below.
>
> - **How robust is this approach if the building changes?**
>
>   - We ran several transfer experiments in which the offline-RL model is learned from the buffer with weather condition A and evaluated on weather condition B (please refer to Table 4 and Figure 8). The results demonstrate that with RUBICON, the performance still is better than the original RBC policy of condition B.
>   - The transfer here is possible without any change to the RL model architecture because the building shares the same state and action space. We have not explored the case where state-action space is different across buildings.
>
> - **Can we see an ablation study where we compare performance of the method to standard offline RL when the rule based policy becomes worse and worse?**
>   - We ran another ablation experiment to observe how the quality of RBC policy affects the performance compared with RUBICON and baseline TD3+BC. We designed two poor quality RBC policies. One is a biased RBC where we jitter the action space in Algo.2. The performance gets worse, but still improves on TD3+BC baseline. The other is to replace RBC with a random policy. In this case, the performance becomes worse than TD3+BC. Results are shown in Figure 15.
>
> - **Since TD3+BC is worse than CQL, as evident from the table, can this method also be applied to CQL to see how well it performs?**
>   - We ran the experiments with CQL+RUBICON, where we integrate RBC policy with CQL instead. The experimental results (Figure 12,13 and Table 8,9) show that CQL+RUBICON improves in performance over CQL. A detailed description is in Appendix C.2. However, overall TD3BC+RUBICON performs the best.
>
> - **Would be good to do some qualitative analysis: when does the RL policy perform particularly much better than the rule-based policy?**
>   - For learning from the expert buffers, RL policy would perform particularly better than rule-based policy. From Table 1, our method RUBICON is outperformed by baselines in most tasks learned from expert buffers. We will add this as a limitation in the paper.
>
> - **What happens if the data composition is different (more representative of real-world systems than random, medium, expert from D4RL)?**
>   - We ran experiments where RUBICON learns from buffers generated from RBC policy. Even when the buffer is generated by RBC policy and the reference policy is RBC itself, our method still outperforms RBC (Figure 11 , Table 7 and description in Appendix C.2).
>   - For real-world application, RUBICON can be used as a transfer learning mechanism where we have a medium buffer from the source building and RBC policy from the target building (Table 4). Such medium buffer data can be obtained from experimental buildings such as those hosted by the Department of Energy (e.g.: [BRITE](https://www.researchgate.net/profile/David-Culler-4/publication/220473428_Reducing_Transient_and_Steady_State_Electricity_Consumption_in_HVAC_Using_Learning-Based_Model-Predictive_Control/links/0deec518b1fc5559c1000000/Reducing-Transient-and-Steady-State-Electricity-Consumption-in-HVAC-Using-Learning-Based-Model-Predictive-Control.pdf) and [FlexLab](https://www.energy.gov/sites/default/files/2014/10/f18/emt80_regnier_042414.pdf)).
>
> - **I encourage authors to reformulate the wording of the paper to be more of an applications paper -- Applying offline RL for building control, and provide benchmarks for RL researchers to iterate on. I think that could be a big win of this paper if done right.**
>   - Thank you for the suggestion. We have changed the wording to make it clear that this is an application paper in the building controls domain. We will open-source our code, baselines, and data.

---

### Decision · Program_Chairs · 2023-01-20

**Decision:**

Reject

**Justification For Why Not Higher Score:**

Ultimately, this is a reasonable application paper.  However, the technical innovation is too limited and the lack of real-world evaluation of a deployed policy makes the evaluation and impact a bit niche for this venue.  I decline to endorse this for acceptance here, but I nevertheless commend the authors for their focus on contributing towards a practical solution for an important application area.

**Justification For Why Not Lower Score:**

N/A

**Metareview: Summary, Strengths And Weaknesses:**

To control HVAC building environments, rule-based control has been the default, but there is an expectation that learning based approaches should outperform them.  The authors introduce a hybrid approach that involves combining a rule-based scheme with learning-based approaches.  In particular, the authors want to improve upon online and offline RL approaches by incorporating a rule-based control policy for regularization.

This is an important application domain, the paper is clear, and the proposed method performs reasonably well empirically, albeit in simulated environments.  Reviewer concerns focus on limited technical novelty and lack of theoretical justification for the approach.